

# On the relationship between tropospheric CO and CO₂ during KORUS-AQ and its role in constraining anthropogenic CO₂

Wenfu Tang[1,2*], Benjamin Gaubert[2], Louisa Emmons[2], Yonghoon Choi[3,4], Joshua P. DiGangi[3], Glenn S. Diskin[3], Xiaomei Xu[5], Cenlin He[6], Helen Worden[2], Simone Tilmes[2], Rebecca Buchholz[2], Hannah S. Halliday[3,7] and Avelino F. Arellano[1]

[1]Department of Hydrology and Atmospheric Sciences, University of Arizona, Tucson, AZ, USA

[2]Atmospheric Chemistry Observations and Modeling Laboratory, National Center for Atmospheric Research, Boulder, CO, USA

[3]NASA Langley Research Center, Hampton, VA, USA

[4]Science Systems and Applications, Inc., Hampton, VA, USA

[5]Department of Earth System Science, University of California, Irvine, CA, USA

[6]Research Applications Laboratory, National Center for Atmospheric Research, Boulder, CO, USA

[7]Now at US Environmental Protection Agency, Research Triangle Park, Durham, NC, USA

Corresponding author: Avelino Arellano (afarellano@email.arizona.edu)


## 1 Abstract

While the complementarity of CO data in monitoring $CO_2$ from fossil-fuel combustion ($ffCO_2$) is
widely known, a rigorous demonstration of its use in reducing uncertainties on top-down regional
$ffCO_2$ emissions is still warranted. Here, we report a case study investigating the regional
covariation of observed and modeled abundances of CO, $CO_2$, and $ffCO_2$ and demonstrating its
implication to joint $CO:CO_2$ inversions. We use data from a recent aircraft field campaign
(KORUS-AQ) conducted over Korea and neighboring regions on May 2016 for this case study.
We use the Community Atmosphere Model with Chemistry (CAM-Chem) to simulate CO, $CO_2$,
$ffCO_2$ and associated source tags, using *a posteriori* fluxes from global $CO_2$ flux inversions and
CO emissions independently calibrated against CO data. Among other model-data comparisons,
CAM-Chem simulations show an underestimation in $CO_2$ (1 ppm), CO (24 ppb) and $ffCO_2$ (1
ppm) against aircraft measurements. These are all within the range of model and data uncertainties.
Although the overall observed enhancement ratio, $\Delta CO/\Delta CO_2$ (~13.3±0.21 ppb/ppm), is well
captured by CAM-Chem (~13.8±0.23 ppb/ppm), we find an overestimation (29 ppb/ppm) for air
samples between 2 to 3 km, where East Asian influence is substantial (35%). The contribution of
$ffCO_2$ from Korea and Japan is smaller (30%) and localized below 3 km, suggesting that regional
$ffCO_2$ and background and non-$ffCO_2$ cannot be neglected in interpreting observed enhancements
in this region. These spatial variations translate in the joint $CO:CO_2$ inversion to increases in *a*
*posteriori* $ffCO_2$ estimates from East Asia (27%±24%) and Korea and Japan (9%±17%). This is
consistent (albeit larger in 1-sigma uncertainty) with our estimate using $^{14}CO_2$ data (27%±9% and
10%±3%, respectively). In contrast, the inversion using only $CO_2$ data shows a decrease by
~5%±27% in East Asia and ~6%±19% in Korea and Japan. Our results show that inversions using
both $CO_2$ and CO can be an effective approach in constraining $ffCO_2$ when the regional variations
of CO and $CO_2$ relationships are appropriately accounted for. Although this further points to the
potential of augmenting current observing system of $CO_2$ with CO for global inverse analyses of
$ffCO_2$ from different regions of the globe, we highlight the need to verify the spatiotemporal
distribution of the covariation of CO with $CO_2$ in both regional and global models. We caution its
use for constraining local $ffCO_2$, unless the spatiotemporal *a priori* flux distribution and surface
processes are reasonably represented, as they may confound the analysis. These have important
implications on inversion studies using columnar data from satellite observations, especially for
regions lacking necessary verification measurements.


## 1.   Introduction

Reducing the uncertainty on top-down estimates of carbon dioxide emissions from fossil fuel combustion ($ffCO_2$) continues to be a challenge. This is due to the dearth of accurate $CO_2$ measurements, including $^{14}CO_2$-derived $ffCO_2$ measurements, with sufficient spatiotemporal coverage necessary to resolve variations in combustion and fuel-use patterns, along with difficulty in teasing out small anthropogenic signature from the large natural sources and sinks dominating the carbon cycle, and the uncertainties in modeling atmospheric transport (e.g., NRC, 2010; Ciais et al. 2014). This challenge remains despite the addition of aircraft and satellite measurements of $CO_2$ abundance in recent years (e.g., Hungershoefer et al. 2010; Chevallier et al., 2014; Houweling et al., 2015). Understandably, global atmospheric $CO_2$ inversions are sharply focused on quantifying land and ocean biospheric sources and sinks because of the significantly larger uncertainties in *a priori* biospheric fluxes of $CO_2$ and transport models than global $ffCO_2$ emissions, together with the reality that current global carbon observing systems have been mostly designed to provide constraints on biospheric fluxes (e.g., Gurney et al., 2003, 2004; Peylin et al., 2013; Schuh et al., 2019). In light of this, global $ffCO_2$ emissions are typically not constrained in these inversions, although their importance has long been pointed out, often in the context of terrestrial and oceanic $CO_2$ flux inversions (e.g., Gurney et al., 2005; Peylin et al., 2011; Saeki and Patra, 2017; Gaubert et al., 2019). As discussed in Andres et al. (2012), the uncertainty in current global bottom-up fossil fuel $CO_2$ ($ffCO_2$) emission inventories is about 10% globally and ranges from a few percent to greater than 50% regionally (or nationally). Combustion activity and efficiency and fuel-use mixtures are still poorly characterized particularly in rapidly developing nations. This is because of the paucity of detailed information on energy use, combustion practices, and pollution control strategies in these regions (e.g., Quillcaille et al., 2018; Andres et al., 2016; Hogue et al., 2016). Most recently, Basu et al. (2020) reported that even at national level, $ffCO_2$ emission inventories in the United States are significantly underestimated by 2 to 3−*sigma* uncertainties. While significant efforts on improving bottom-up $ffCO_2$ emission inventories through detailed accounting on other sources of information have been made (e.g., Gurney et al. 2009; Rayner et al., 2010; Asefi-Najafabady et al., 2014), the lack of finer scale measurements to verify these inventories remains to be addressed. Recent reports have recognized this limitation and recommend augmenting the current observing system with systems that can help identify dynamical, physical, and chemical signatures of $ffCO_2$ at regional scales (e.g., Ciais et al., 2015).

Observational constraints on $ffCO_2$ from radioactive tracer $^{14}C$ have largely been established. Measurements of $^{14}C$ of $CO_2$ are able to separate the fossil and biogenic contributions in observed $CO_2$ and serve as useful tracer of $ffCO_2$ emissions from different regions (e.g., Levin et al., 2003, 2008; Turnbull et al., 2006, 2009, 2011, 2015; Graven et al., 2009, 2018; Miller et al. 2012; Basu et al., 2016; Niu et al., 2016; Berhanu et al., 2017; Nathan et al., 2018; Basu et al., 2020). Fossil C is $^{14}C$ free due to the much shorter half-life of $^{14}C$ (~5,700 years) than the age of the fossil (~/>$10^6$ years). Measurements of radiocarbon content ($^{14}C$) of atmospheric $CO_2$ may sensitively indicate fossil fuel $CO_2$ (delta value of -1000 ‰) additions to the air sample, showing $^{14}C$ is lower than contemporary background (clean air) $\Delta^{14}C$ values. While extremely useful, such measurements have only been made routinely in few locations around the world (e.g., Turnbull et al., 2007 at some Global Atmospheric Watch (GAW) sites; Berthanu et al., 2017 in Switzerland) or occasionally at specific region during field measurement campaigns (e.g., Turnbull et al., 2012 during INFLUX: Indianapolis Flux Experiment). These are still limited in providing sufficient top-down constraints on $ffCO_2$ emissions at a regional to global scale, especially emissions from poorly





observed developing regions of the world. The utility of these measurements in constraining $ffCO_2$,
in the context of deploying a potential network of these measurements within a joint inversion
framework, have recently been assessed, albeit only through observing system simulations
experiments or OSSEs (e.g., Basu et al., 2016; Nathan et al. 2018; Wu et al., 2018; Wang et al.,
2018). The recent study by Basu et al. (2020) using real $\Delta^{14}CO_2$ from NOAA sites in the United
States, however, shows very promising results on $ffCO_2$ emission constraints at national level. A
consistent finding among these studies is that, while there is strong potential to reduce national
$ffCO_2$ uncertainties (1% on yearly basis to 5-10% on monthly basis), such atmospheric-based
approach to estimate emissions also requires careful consideration of errors in transport, systematic
bias and accuracy of measurements, and characterization of background $CO_2$.
It is particularly appealing to consider synergies between $CO_2$ and $ffCO_2$ and air quality monitoring
(AQ) observations (e.g., CO, $NO_2$), since in an environment where combustion activities are
dominant, these species being monitored regularly often share the same dominant source category.
Both are co-emitted during carbonaceous-fuel (fossil fuel-FF or biofuel-BF) generation,
combustion, and distribution processes. In particular, CO is produced when combustion is
incomplete; otherwise carbon in the fuel is oxidized to $CO_2$ at equilibrium levels of CO. And so,
observing the relative abundance of $ffCO_2$, $CO_2$, and CO in this environment should provide useful
synergistic information on their associated combustion-related emissions. This is the case for CO,
for which larger number of observations are available from ground network, airborne, and satellite-
derived measurements. Such datasets have been utilized in the past to provide additional
constraints on combustion-related emission patterns in urban regions and biomass burning
activities at local to global scales. They have been extended to provide insights on $ffCO_2$ or fire
$CO_2$ (e.g., Suntharalingam et al., 2004; Palmer et al. 2006; Wang et al., 2010; Turnbull et al., 2011;
Brioude et al., 2013; Lopez et al., 2013; Silva et al., 2013; Konovalov et al., 2014; Lindenmaier et
al. 2014; Ammoura et al., 2016; Bowman et al., 2017; Super, 2018; Nathan et al., 2018; Boschetti
et al., 2018 among others), as well as to identify and characterize air masses (e.g., Halliday et al.
2019). These studies used CO as an indirect tracer of combustion through a variety of ways: data
analysis, model-data comparison, modeling, or inversions at different scales and region depending
on their application. $ffCO_2$ emissions in bottom-up emission inventories are calculated using
information on combustion activity, emission factor, and combustion efficiency (CE). A typical
indicator of combustion efficiency is the ratio of measured $CO_2$ to (CO + $CO_2$). Differences in CE
across different source sectors (e.g., power plant: high CE, domestic heating: low CE, flaming fire:
high CE, smoldering fire: low CE) can be distinguished with atmospheric measurements of CO
and $CO_2$. In particular, derived CO:$CO_2$ enhancement ratios near a source region are used to verify
bulk CO:$CO_2$ emission ratios from these inventories. Hourly $ffCO_2$ emission profile from traffic
are also deduced from measurements of CO (e.g., Vogel et al., 2010; Super, 2018). Because of its
medium-length lifetime (1 to 2 months), CO is also a useful tracer of pollution (incl. $ffCO_2$)
transport. Tracking urban plumes using CO can help enhance horizontal and vertical transport
signatures of $ffCO_2$ plumes, which may be difficult with $CO_2$ measurements alone due to its longer
lifetime and the influence of a large biospheric signal.
From a spatiotemporal sampling standpoint, these CO datasets are strongly complementary,
especially in the absence or lack of $CO_2$ and $^{14}CO_2$ measurements. In addition, identifiable
physico-chemical constraints from CO on anthropogenic $CO_2$ emissions and their transformations
can also be exploited (i.e., oxidation of reduced carbon to $CO_2$, Suntharalingam et al., 2005; Nassar
et al., 2010; Wang et al, 2020). In fact, the recent study by Wang et al. (2020) highlighted the


impact of accounting for the chemical production of $CO_2$ on estimates of global carbon sinks. Yet,
unlike $^{14}CO_2$, these types of information from CO are generally confounded by: a) sharp
differences in their associated sinks (through chemical transformation) downwind of its source;
and hence differences in lifetimes and background concentrations across space and time; b)
biogenic sources even within an urban environment; and c) variations in the effectiveness of
pollution control strategies for CO between sectors within an urban region. Note that many of these
confounding factors become more dominant at finer scales of the study region. Hence, constraints
on $ffCO_2$ emissions from CO data has to be exploited at appropriate scales. The joint inversion of
CO and $CO_2$ by Palmer et al. (2006), for example, clearly shows that estimates of anthropogenic
$CO_2$ can be very sensitive to assumptions of the relationship between CO and $CO_2$, which can then
also influence the accuracy of biospheric flux estimates. Due to these factors, its use in constraining
regional to global $ffCO_2$ emissions remains to be limited, despite its complementarity and the
availability of a large number of its measurements. In our view, it is critical to first understand and
better characterize the observed and modeled relationship between CO and $CO_2$ abundance before
incorporating such information at appropriate scales in systems directed towards improving our
capability to attribute the sources of $ffCO_2$.

## 1.1 Objectives

The main goals of this study are to assess the relationship between CO and $CO_2$ that can be inferred
from observations and a climate-chemistry model and demonstrate its implications to joint
$CO:CO_2$ inversion. Here, we take advantage of $^{14}CO_2$, $CO_2$, and CO measurements during a recent
field campaign conducted over Korea on May 1-June 10, 2016. This study is a continuation of our
work on evaluating the Copernicus Atmosphere Monitoring Service (CAMS) CO and $CO_2$ high
resolution forecast and analysis products during Korea-United States Air Quality (KORUS-AQ)
field campaign (Tang et al., 2018). This also serves as a complementary study to our recent work
on quantifying the source contributions of CO over Seoul during KORUS-AQ using regional tags
or tracers in the Community Atmosphere Model with Chemistry or CAM-chem (Tang et al.,
2019a), and to the study by Halliday et al. (2019) on characterizing air masses using short-term
$CO:CO_2$ ratios during the same field campaign.
The specific objectives of this study are three-fold: 1) We introduce and evaluate a single-model
analysis framework for multi-species analysis and inversions; 2) We examine the modeled and
observed spatial distribution of the inferred relationship between CO, $ffCO_2$ and $CO_2$; and 3) We
demonstrate the role of CO in refining observational constraints in regional $ffCO_2$ emissions
through Bayesian synthesis inversions. This framework is directed towards simulating the
abundance of CO and $CO_2$ in CAM-chem, based on observationally constrained surface fluxes for
$CO_2$ from global flux inversions and a 'best emission scenario' for CO from our previous work. In
addition, we added a capability in CAM-Chem to tag the regional sources of ffCO and $ffCO_2$,
which we could not do in our previous study using the CAMS operational forecasting system.
These tags enable us to assess the relationship of regional $ffCO_2$ and $CO_2$ which would not be
possible in this type of model using observations of $ffCO_2$ and $CO_2$ alone. We note that the system
approach we are suggesting in this work is similar to previous global studies of these species,
particularly with Palmer et al. (2006), which also considered aircraft measurements from a field
campaign conducted in 2001 over similar (albeit larger) region (TRACE-P: TRansport and
Chemical Evolution over the Pacific, Jacob et al., 2003). We view this work to be complementary
to their study by updating the state of $CO:CO_2$ ratios in this region after 15 years. We emphasize



that our focus, however, is to characterize these ratios in the context of refining ffCO$_2$ constraints,
and not purely in optimizing global flux inversions. The main difference in modeling framework
between this work and previous studies is the use of *a posteriori* fluxes (and emissions), rather
than *a priori* fluxes in simulating the abundance. Also, while Halliday et al. (2019) and Tang et al.
(2018) have already presented such characterization of CO:CO$_2$ ratios during KORUS-AQ, this
study is unique in a way that we use the tagged ffCO$_2$ component of this system to attribute the
contributions of regional ffCO$_2$ on these ratios.
This paper is structured as follows. In Section 2, we describe the model and datasets used in this
study. In Section 3, we evaluate the modeled CO, CO$_2$, and ffCO$_2$ during KORUS-AQ. We
characterize the spatial distribution of CO and CO$_2$ relationships and its implication to CO:CO$_2$
inversion in Sections 4 and 5, respectively. We present the discussion and general implications of
this study in Section 6 and our conclusions in Section 7.
**2.      Methods and data description**
**2.1      CESM/CAM-Chem**
The Community Earth System Model version 2 (CESM2) includes atmosphere, land, ocean, land
ice, sea ice, and river components, all of which are connected by a coupler (Danabasoglu et al.,
2020). CAM-chem is the atmospheric chemistry component of CESM, coupled with the land
model (Lamarque et al., 2012). In CESM2, CAM-chem includes a significantly updated
tropospheric chemistry mechanism (MOZART-T1), coupled to a VBS (volatility basis set) scheme
for the formation of Secondary Organic Aerosols (SOA), allowing to simulate explicitly the
tropospheric and stratospheric composition (Emmons et al., 2020; Tilmes et al., 2019).
**2.1.1.    CO$_2$ fluxes and CO emissions**
The default CAM-chem configuration for greenhouse gases (CO$_2$ and CH$_4$) simulations are carried
out by prescribing mixing ratios of these species at the model surface layer, following the CMIP6
protocol (Meinshausen et al., 2017). The CO$_2$ mixing ratios at the surface layer are based on
zonally averaged observed CO$_2$ from NOAA ESRL Carbon Cycle Cooperative Global Air
Sampling Network (Dlugonkencky et al., 2015). In this study, however, we simulate atmospheric
CO$_2$ explicitly with an ensemble of external CO$_2$ fluxes. Specifically, we use the *a posteriori* fluxes
from CAMS Greenhouse Gases (GHG) flux inversion (CAMSv17r1; Chevallier et al., 2005, 2010,
2013, 2018), CarbonTracker 2017 (CT2017; Peters et al., 2007, with updates documented at
http://carbontracker.noaa.gov), and CarbonTracker Europe 2018 (CTE2018; van der Laan-Luijkx
et al., 2017). CarbonTracker is a global modeling system of CO$_2$ developed by NOAA with a
nested grid on North America (Peters et al., 2007, with updates documented at
http://carbontracker.noaa.gov). CarbonTracker Europe is developed based on CarbonTracker (van
der Laan-Luijkx et al., 2017). Both CT2017 and CTE2018 provide fluxes of fossil fuel, fire, land,
and ocean components, which we use for our tagging of regional sources of ffCO$_2$. CAMSv17r1
is produced by the inversion system called PyVAR (Chevallier, 2018). We regridded all these CO$_2$
fluxes to match our CAM-chem resolution (0.95°×1.25°). Details of the fluxes are listed in Table
1 (and Table S1). Technical details for simulating atmospheric CO$_2$ explicitly with external CO$_2$
fluxes in CAM-chem are included in the supplementary material (Text S1). The term "CAM-chem
CO$_2$", "simulated CO$_2$" and "modeled CO$_2$" in this study stand for the atmospheric CO$_2$ simulated



with the aforementioned method rather than the atmospheric $CO_2$ prescribed in CAM-chem by
default, unless stated otherwise. To simulate CO in CAM-chem, we use the Fire INventory from
NCAR (FINN; Wiedinmyer et al., 2011) for biomass burning CO (as well as other related species
such as NMVOCs) emissions, and the Hemispheric Transport of Air Pollution version 2 inventory
(HTAPv2; Janssens-Maenhout et al., 2015) for anthropogenic CO (as well as other related species
such as NMVOCs) emissions. In our previous evaluation of CO (Tang et al., 2019a), we calibrated
these HTAPv2 emissions by doubling its associated CO and VOC emissions in East Asia and
Korea to match the CO data in the region. The CT2017 $CO_2$ fluxes and CO emissions are shown
in Figure 2 while the other 3 $CO_2$ fluxes and the ensemble standard deviation are shown in Figure
S1.

### 217 2.1.2 Implementation

We run four CAM-chem simulations with simulated $CO_2$ as well as full chemistry (e.g., CO, $O_3$)
for the year 2016, using four sets of $CO_2$ fluxes as described in Table S1 (including CT2017 3-
hourly fluxes, CT2017 monthly fluxes, CTE2018 fluxes, and CAMS fluxes). We run CAM-chem
with the model meteorological fields nudged towards Modern-Era Retrospective analysis for
Research and Applications, Version 2 (MERRA-2, Gelaro et al., 2017). The CAM-chem $CO_2$ is
initialized with CT2017 mole fraction fields on January 1st, 2016, while other variables in CAM-
chem (e.g., CO) are initialized with results from previous CAM-chem simulations. The associated
global budgets for our CO and $CO_2$ simulations are presented in Table S2. We also show in the
supplementary material (Figure S2) the corresponding global $CO_2$ abundance for each flux product
that we used and the concentration fields from CT2017. This is intended to ensure that: a) CAM-
chem reasonably reproduces the CT2017 $CO_2$ fields when using CT2017 fluxes; b) appropriate
accounting of each tag is carried out; and c) mass is conserved. Overall, our simulation results
produce $CO_2$ fields comparable to current $CO_2$ analyses while carbon is reasonably accounted for.
Differences in $CO_2$ mass is ~0.001% of initial burden which may be attributed to a cutoff of model
top at ~2 hPa. In most of our analysis, we will use CAM-chem with CT3h fluxes as our base
simulation. Comparisons of simulated $CO_2$ between other fluxes are only intended to show the
total spread and not necessarily to draw conclusions on emissions or performance of these fluxes
since these fluxes vary in spatiotemporal resolution.

### 237 2.1.3 Tagging ffCO$_2$ and CO

As previously noted, we developed a capability in CAM-chem to tag different source regions
and/or emission types for $ffCO_2$ in addition to the existing CO tagging mechanism. This tagging
approach is further described in Appendix A. We run one tagged simulation for May to June 2016
(the KORUS-AQ campaign period) using the same model configuration but only with $CO_2$ fluxes
and CO emissions from the tagged regions defined in Figure 1. Note that for this particular
simulation, we use CT2017 3-hourly fluxes for $CO_2$ (CT3h) and a relatively well performing CO
emission scenario from Tang et al. (2019a) which is based on HTAPv2 for anthropogenic CO
emissions. We tag $ffCO_2$ from 11 regions in East Asia (shown in Figure 1) with one additional tag
that accounts for fossil fuel emissions from the rest of the world (ROW), to complete the $ffCO_2$
budget in CAM-chem. The $CO_2$ and CO tags are initialized with zero fields on Jan 1, 2016, so that
only the emissions in 2016 are accounted when analyzing the relationships between $ffCO_2$ tags,
CO and $CO_2$. Note that this Eulerian tagging method will be used to account for the relative
contribution of different source regions to modeled $CO_2$. This is similar, in principle, to forward





and backward Lagrangian trajectory models of air parcels like FLEXible PARTicle dispersion
model (FLEXPART, Stohl et al. 2009) used in Turnbull et al. (2011).

## 2.2 Observational datasets

While we focus our analysis on KORUS-AQ measurements, we also use other datasets to assess
the overall consistency of simulated CO and $CO_2$ (incl. their relationships) during this period.
Please see the supplementary material (Table S3) for more information.

### 2.2.1 Aircraft measurements of CO, $CO_2$, and ff$CO_2$ during KORUS-AQ

The Korea United States Air Quality (KORUS-AQ) field campaign was conducted over South
Korea and its surrounding waters from May to June 2016 (Al-Saadi et al., 2014; https://www-
air.larc.nasa.gov/missions/korus-aq/). The flight tracks are shown in Figure 1. The Atmospheric
Vertical Observations of $CO_2$ in the Earth's Troposphere (AVOCET; Vay et al., 2011) and
Differential Absorption CO Measurement (DACOM; Sachse et al., 1987, 1991) were onboard the
NASA DC-8 aircraft to measure $CO_2$ and CO, respectively. AVOCET uses a modified LI-COR
6252 instrument with time response of 1 second, precision and accuracy of 0.25 ppm (Vay et al.
2003). The DACOM instrument has a time response of 1 second, precision of 0.4 ppb and accuracy
of 2%. These instruments were calibrated in flight during the campaign with standards from
NOAA ESRL traceable to WMO $CO_2$_X2007 (Zhao & Tans, 2006) and CO_X2014A (NOAA,
268 2020).

In addition, 46 radiocarbon ($^{14}CO_2$) samples have also been collected onboard the NASA DC-8
aircraft during KORUS-AQ campaign with WAS (Whole Air Sampler team at UCI) flask samples
and measured at W.M. Keck Carbon Cycle Accelerator Mass Spectrometer lab at UC, Irvine.
ff$CO_2$ calculation from $^{14}C$ of $CO_2$ followed the approach by Turnbull et al. (2011), Miller et al.
(2012), and Lehman et al. (2013). In particular, we use Eq. 1 of Turnbull et al. (2011) to derive
$CO_{2ff}$ (using their notation) with a background value of $\Delta^{14}CO_2$ (or $\Delta_{bg}$ in their notation) of 15‰.
This value is adopted based on $\Delta^{14}CO_2$ data in Point Barrow, AK (13.9±1.5 ‰) and Niwot Ridge,
CO (NWR, ~15 ‰) during the same May-June 2016 period corresponding to the KORUS-AQ
campaign. This choice follows in the same manner to the discussion in Turnbull et al. (2011) on
representative background values. As they pointed out, the high-altitude clean air sites, like NWR,
appear to be representative of Northern Hemisphere midlatitude background and similar to
Jungfraujoch, Switzerland which was also previously used in other studies to represent the
background. They also pointed out that differences on the choice of background values do not
significantly affect their results since these differences are smaller than the enhancements in their
study region (Tae-Ahn Peninsula, Korea -TAP, Shangdianzi, China - SDZ), which is similar to our
study region. In fact, we find that $\Delta^{14}CO_2$ during the campaign are always lower than 15‰. For
the correction of the other effects, such as heterotrophic respiration and biomass burning (see 2nd
term of Eq 1 in Turnbull et al., 2011, bias β in Eq 4 of Turnbull et al., 2009), we use -0.5 ppm
corresponding to their estimate of this correction for summer months. We also follow a similar
reasoning regarding the relatively small (with some that are not quantifiable in North Korea nuclear
facility) $^{14}C$ influence on emissions of ff$CO_2$ from nuclear powerplant activities in Korea, since all
the powerplant sites are using pressurized water reactor ((https://www.world-
nuclear.org/information-library/country-profiles/countries-o-s/south-korea.aspx). In the same



manner, the 1-*sigma* uncertainties in ffCO₂ and $\Delta^{14}CO_2$ are estimated to be 1 ppm and ±1.8‰,
respectively.

### 2.2.2    Satellite-derived measurements of CO and CO₂

To provide a broader spatial context, we use retrievals of $CO_2$ column-averaged dry-air mole
fraction ($XCO_2$) from the NASA Orbiting Carbon Observatory-2 (OCO-2), version 8, level 2 (L2)
Lite product with the recommended quality flag (i.e., xco2 quality flag equals to 0) (Boesch et al.,
2011; Osterman et al., 2017; O'Dell et al., 2018). The uncertainty of $XCO_2$ retrievals is about 1-2
ppm (Wunch et al., 2017). For CO, we use total column retrievals (XCO) of the Measurements Of
Pollution In The Troposphere onboard Terra, version 7, Level 2, multispectral (thermal
infrared/near infrared; TIR/NIR) (MOP02J, L2, V7) with the recommended quality flag (i.e.: cloud
mask from MOPITT and Moderate Resolution Imaging Spectroradiometer agree on clear for
Cloud Description; sum of Retrieval Anomaly Diagnostics equals to 0; solar zenith angle is less
than 80) (Worden et al., 2010; Deeter et al., 2017). The model equivalent is calculated by first
interpolating the model profile to the location of the satellite retrieval and applying the associated
*a priori* profile and averaging kernel.

### 3.    Comparison of modeled and observed CO, CO₂, and ffCO₂

A comprehensive summary of our comparison against KORUS-AQ (and other types of observing
platforms) is presented in Table 2 (and Table S4). Overall, these simulations show relatively good
agreement. The error statistics are comparable with state-of-the-art $CO_2$ and CO model
simulations. The $CO_2$ simulations, in particular, closely matches with CT2017 mole fractions. The
bias in modeled $CO_2$ against observations are also within the range of biases in other models. For
example, the bias in CAM-chem against TCCON Saga site (Shiomi et al., 2017) range from -0.6
to -1.5 ppm, which is within the error range of OCO-2 MIP $CO_2$ (Crowell et al. 2019) for the same
period. We emphasize here that the statistics of such comparisons (including error statistics like
bias, root-mean-squared-error, and correlation) are estimated for instantaneous data points during
the KORUS-AQ period (May to June 2016) or only for a single year in 2016 (in the case of NOAA
and TCCON comparisons, see Figure S3 and S4). This period corresponds mostly to the peak in
global average $CO_2$ in 2016 (Figure S2). Error comparison with other models should be limited to
this specific month and year.
As shown in Figure 2, the mean spatial covariation of major sources of $CO_2$ and CO in the region
(Beijing, Shanghai, Guangzhou, Seoul, Tokyo) for this period are broadly similar. However, they
are more pronounced in observed $XCO_2$ than XCO. We attribute this to relatively lower sensitivity
of MOPITT retrievals near the surface and differences in the source magnitudes between large
cities in East Asia and Korea and Japan. While the overall correlation (R=0.46-0.68) and bias (~0.5
to 0.8 ppm) between modeled and observed $XCO_2$ are relatively moderate, the modeled $XCO_2$ is
slightly underestimated in source regions (e.g., Beijing, Tokyo, Seoul) and overestimated in the
Yellow Sea and northern latitudes. The modeled XCO, on the other hand, appears to be
overestimated across the East Asian domain (i.e., R=0.76, bias~6.4 ppb) with higher variability
(27 ppb) than observed (19 ppb). This is most likely due to the previous scaling (doubling) of
anthropogenic CO and VOC HTAPv2 emissions in East Asia and Korea, as well as possible
overestimation of fires in the region from FINN. Observed "background" of $XCO_2$ (401.75 ppm)
and XCO (80.01 ppb) are slightly overestimated (402.94 ppm) and underestimated (79.50 ppb) by





CAM-chem. "Background" is broadly defined here as 5th percentile across the domain for the
May 2016 period. On the other hand, the 95th percentile of observed $XCO_2$ (408 ppm) and XCO
(137 ppb), broadly representing "polluted" conditions, are underestimated (407 ppm) and
overestimated (156 ppb) by CAM-Chem suggesting variations in overall bias between
"background" and "polluted" conditions in this region.

### 3.1 Comparison against KORUS-AQ CO and $CO_2$ measurements

Similar to Tang et al. (2018), we organized these aircraft measurements into five flight groups to
facilitate a more detailed comparison of the spatial distribution of CO and $CO_2$ in the region. These
groups represent variations in sampling of air mass characteristics during the campaign (see Figure
3f). In particular, the Seoul flight group represents air samples over Seoul, which is characterized
to have a dominant signature from anthropogenic combustion processes, while Taehwa represents
air samples that may have both biospheric (nearby forest) and anthropogenic (Seoul metropolitan)
influence. The flights over the West Sea were designed to capture China pollution outflows by
conducting only on days when a China outflow is expected to be present. The Seoul–Jeju flight
group represents air samples over local power plants, transported air from the West Sea, and over
nearby croplands, while the Seoul–Busan flight group represents air samples over forest, rural, and
Busan urban regions.
We show in Figures 3 and 4 the average horizontal and vertical distribution of observed and
modeled CO and $CO_2$ for different flight groups. The overall statistics, which are calculated across
all data points within a flight group, are also summarized in Table 2. For comparison with CAM-
Chem $CO_2$, we also show the model equivalent $CO_2$ from the mole fractions reported in CT2017
system, which uses a different transport model (TM5). It is evident from these comparisons, that
while the spatial gradients in observed $CO_2$ are relatively captured by CAM-chem (albeit also
showing lower variability than observed), there appears to be a low negative bias (i.e., model minus
obs) in nearby source regions (Seoul and its west coast), and over West Sea. The range of observed
$CO_2$ values across flight groups, altitude, and KORUS-AQ period starts from a low of 408 ppm
(Taehwa) to a high 415 ppm (Seoul) with the standard deviation ranging from 4 ppm (Seoul-
Busan) to 13 ppm (Seoul). The model equivalents are slightly lower and less variable: 408 ppm
(Taehwa) to 412 (Seoul) with standard deviation between 3.5 ppm (West Sea) and 10.5 ppm
(Seoul). Such a slight underestimation is shown to occur in the lowermost layer of the observed
$CO_2$ vertical profiles (Figure 4) where the median bias and interquartile range (IQR) across flight
groups is -2.7±4.6 ppm. Yet over the southern coast of Korean peninsula, as well as the transect
from Seoul to Busan, there is a positive bias. A slight overestimation can also be seen in the air
aloft (Taehwa, Seoul-Busan, and Seoul-Jeju), where the median bias and IQR is 0.6±0.6 ppm.
Above 3 km, the 5th percentile of $CO_2$ data (All flights) is 403.5 ppm, while its model equivalent
is 405.1 ppm. Such underestimation and overestimation are consistent with our comparison against
OCO-2 $XCO_2$ indicating variations on the influence of local and regional "pollution"
(underestimation) and "background" (slight overestimation) on these biases. Differences between
CAM-chem and CT2017 $CO_2$ are small except in below 2 km. The median difference in bias
between CAM-chem and CT2017 across flight groups and altitude is -0.1±0.6 ppm, where much
of the variability comes from West Sea. Since both systems use the same flux distribution (CT3h),
we mostly attribute this difference to the coarser resolution ($3^o$ x $2^o$) of the CT2017 mole fraction
fields that we obtained from Carbon Tracker, which may not be able to better represent local
variations in $CO_2$. It is quite possible that these differences are due to differences in boundary layer



representation due to coarser vertical resolution and/or different treatment of boundary layer
processes between TM5 and CAM-chem. The overall bias in CAM-chem (-1 ppm) is also
comparable (albeit opposite in sign) to the bias in CAMS forecast and analysis system (0.8 to 2.2
ppm) that we reported in Tang et al. (2018). This system is based on the Integrated Forecasting
System (IFS) of the European Centre for Medium-Range Weather Forecasts (ECMWF) combined
with modules for atmosphere composition (Flemming et al., 2017, Agustí-Panareda et al., 2017),
biospheric $CO_2$ fluxes from terrestrial vegetation (Boussetta et al., 2013), four-dimensional
variational data assimilation (Inness et al., 2019), and biogenic flux adjustment (Agustí-Panareda
et al., 2016). Note that the $CO_2$ fluxes in this system are different from GHG CAMSv17r1
(Chevallier, 2018), which we used as one of *a posteriori* $CO_2$ fluxes in model. Unlike in CAM-
chem, where we see an underestimation of $CO_2$ in the boundary layer, the positive bias in CAMS
is systematic across the vertical profiles for all flight groups, except over West Sea (see Figure 4
of Tang et al. 2018).
In contrast to our comparison with MOPITT XCO across East Asian domain, the modeled CO
over Korea during KORUS-AQ is generally underestimated (model minus obs: -20 to -35 ppb),
except over the west of Seoul and southern Korea. The range of observed CO values across flight
groups, altitude and KORUS-AQ period starts from a low of 163 ppb (Taehwa) to a high of 266
ppb (Seoul) with the standard deviation ranging from 64 ppb (Seoul-Busan) to 143 ppb (West Sea).
The model equivalents are lower and less variable: 143 ppb (Taehwa) to 237 (Seoul) with standard
deviation between 62 ppb (Seoul-Busan) and 133 ppm (Seoul). This is reflected in the CO vertical
profiles, where across most of flight groups (except Seoul-Busan) the modeled CO is
underestimated below 2 km (median bias and IQR across flight groups is -41 ±24 ppb) and above
3 km (-12 ±13 ppb). The only overestimation in modeled CO (median bias of +3 ppb), which is
also reflected in the higher variability of the bias (IQR=84 ppb), can be found at 2-3 km aloft over
Seoul (80 ppb), West Sea (67 ppb) and at 4-5 km over Seoul-Jeju (32 ppb). Above 3 km, the 5th
percentile in observed and modeled CO are 97 and 86 ppb, respectively. Below 3 km, similar
negative bias of ~12 ppb (420 ppb versus 432 ppb) can be found. This suggests an underestimation
of CO in "background" conditions by CAM-chem across the vertical profile in the KORUS-AQ
sampling domain. The regional influence at 2-3 km, on the other hand, is overestimated, as is also
reflected in MOPITT XCO, which we attributed to an overestimation of "polluted" conditions in
the model. The overall negative (and systematic) bias in CO is attributed to an underestimation of
secondary and background CO or an overestimation of OH, since we still see an underestimation
despite previous scaling of East Asia's and Korea's anthropogenic CO and VOC emissions. We
expect that anthropogenic sources of CO in this region is already overestimated. This systematic
bias has been reported in Tang et al. (2019a), which implies considering optimizing secondary CO
and indirectly constraining CO loss due to OH together with primary CO emissions (Gaubert et
al., 2020). Relative to CAMS CO, the overall mean bias against KORUS-AQ in CAM-chem (-24
ppb) is also comparable to CAMS (-20 to -25 ppb). Note that the CAMS system assimilates
MOPITT XCO among other datasets into their forecasting system.
The correlations between $CO_2$ and CO errors (bias) are relatively moderate across all flight groups.
These error correlations range from 0.36 over Seoul to 0.57 over West Sea, and 0.40 over All
flights. These are lower than CAMS CO and $CO_2$ forecasts and analysis (i.e., 0.64-0.90 over Seoul,
0.80-0.82 over West Sea, and 0.49 -0.61 overall). Since $CO_2$ and CO simulations share a common
transport in CAM-chem, lower error correlation in CAM-chem can be due to larger inconsistencies
in representing $CO_2$ and CO sources and sinks in this model. And since both CO and $CO_2$



simulations are consistently underestimating surface concentrations while the same set of
simulations underestimate and overestimate concentrations aloft, respectively, this suggests that
biases in regional sources and sinks are inconsistent between CO and $CO_2$. Although this
inconsistency is expected by design since we used emissions and fluxes from different inventories
and analysis system to highlight variations and potential errors in effective emission ratios, this
also implies the need for accounting for these errors within a multi-species optimization approach.
**3.2    Comparison against KORUS-AQ $^{14}CO_2$-derived ff$CO_2$ measurements**
Figure 5 shows the horizontal (5a), vertical (5c), and temporal distributions (5d) of $^{14}CO_2$
measurements during the campaign. Sample IDs are indicated in the sample location along with
approximate time stamps for a group of samples. We compare these with model ff$CO_2$, which is
calculated as the sum of ff$CO_2$ abundance from the 12 tagged ff$CO_2$ emissions. We note that the
model ff$CO_2$ is not exactly the same as ff$CO_2$ derived from the $^{14}CO_2$ measurements because of
our assumption of initial condition (accounting for emissions from January 1, 2016). As described
in section 2, ff$CO_2$ is derived from $^{14}CO_2$ using a $\Delta^{14}CO_2$ background value representative of the
entire KORUS-AQ campaign period. Since these airborne measurements are taken close to the
fossil fuel emission sources, and hence the variations in the ff$CO_2$ (accumulated since Jan $1^{st}$,
2016) are expected to mostly capture the spatial and temporal variations of regional ff$CO_2$ derived
from $^{14}CO_2$ measurements. We expect that the tagged ff$CO_2$ outside of this region is small and can
be lumped as an offset in ff$CO_2$ initial condition. Figure 5b also shows a scatter plot of ff$CO_2$
derived from the $^{14}CO_2$ measurements and ff$CO_2$ from CAM-chem. We note that there is a lack of
variability in the model for low ff$CO_2$ samples (model standard deviation of 8.6 ppm), as shown
by points clustering around 9 ppm by the model, in contrast to 1 to 12 ppm by the data (obs standard
deviation of 13.2 ppm). This may be related to the relative coarse model resolution ($0.9° \times 1.25°$).
Despite the lack of variability in the model and the limited $^{14}CO_2$ samples, the overall correlation
between ff$CO_2$ derived from $^{14}CO_2$ measurements and modeled ff$CO_2$ tags is moderate (R=0.51).
We identified five (5) data points where derived ff$CO_2$ is significantly high (or low) relative to
their model equivalents (i.e., >90$^{th}$ percentile of the variance of residual). These points are marked
as red (orange) points in Figure 5b. Without these five data points, derived ff$CO_2$ and modeled
ff$CO_2$ have a better correlation of 0.82 ($R^2$=0.67), which is significant at >99% confidence interval.
Note that the average $^{14}CO_2$ values for this campaign period (May 2016), excluding these 5 points,
is $13.2 \pm 9.5$ ppm, while the $10^{th}$ and $90^{th}$ percentiles are in the order of 4.3 and 26.1 ppm,
respectively. This is relatively consistent (albeit higher) with the values from Turnbull et al. (2011)
at Tae-Ahn Peninsula (NOAA/TAP is west coast of Seoul), where the average $CO_2$ff they reported
is $8.5 \pm 8.6$ ppm and 0.4 and 23.2 ppm for $10^{th}$ and $90^{th}$ percentile across a different period (~2005-
2010). The recent study by Lee et al. (2020) at Anmyeon-do (NOAA/KMA-GAW/AMY is 24 km
away from TAP) reports a mean value of $9.7 \pm 7.9$ ppm (with a range between -0.05 to 32.7 ppm)
for the more recent period from May 2014 to May 2016. The value of ff$CO_2$ derived from
interpolated values of NOAA/KMA-GAW/AMY $CO_2$ (417 ppm) and $\Delta^{14}CO_2$ (-15‰) fitted curves
(https://www.esrl.noaa.gov/gmd/dv/iadv) is roughly around 11.8 ppm using the same assumptions
of $\Delta^{14}CO_2$ in the region. We find a relatively higher value during KORUS-AQ as there are more
polluted air masses sampled over Seoul and West Sea during the campaign. These relatively higher
values imply a slight increase in derived ff$CO_2$ in this region. This is reflected in the trend of the



fitted curves for $CO_2$ (increasing) and $\Delta^{14}CO_2$ (decreasing) at AMY and consistent with the
analysis by Lee et al. (2020).
The regional contributions to modeled ff$CO_2$ are superimposed in the bar plots of Figure 5d. The
observed and modeled $CO_2$ and CO corresponding to the same air samples are also shown in Figure
5d to show the relationship between $CO_2$, CO, and ff$CO_2$. While we will discuss this in more detail
in the next two sections, we introduce these tags to point out that the main contributors to the
modeled ff$CO_2$ during the campaign are the nearby source regions in East Asia and Korea. ff$CO_2$
ROW has relatively flat contribution across all samples. Including an offset of 1 ppm to account
for errors in initial condition, the model exhibits a low bias of 1 ppm compared to derived ff$CO_2$.
Note that ff$CO_2$ only accounts a small fraction of observed $CO_2$, even near large source regions
like Seoul. We also note that the 2 sample points over Seoul, where the modeled ff$CO_2$ is
significantly overestimated, correspond to large overestimation in CO when East Asia has
relatively moderate contribution and overestimation in $CO_2$ when Korea's contribution is expected
to be dominant. On the other hand, the 3 sample points over the west of Seoul and West Sea, where
modeled ff$CO_2$ is significantly underestimated, correspond to an underestimation of CO and $CO_2$
regardless of the main source contributor. Again, this variation is consistent with the variation in
bias in "polluted" conditions of modeled CO and $CO_2$ in East Asia described earlier. We attribute
these  differences to the following: (1) errors in initial condition of ff$CO_2$; (2) $CO_2$ (and CO) FF/BF
emissions used in this study may be underestimated (overestimated) over East Asia and Korea;
and (3) the vertical mixing may be overestimated by CAM-chem. We will further investigate these
differences in section 5, where we conducted an inversion using derived ff$CO_2$.
**4.        Observed and modeled relationships of CO and $CO_2$**
In this section, we present a closer look at the variations in CO:$CO_2$ correlation ($R_{CO,CO_2}$) and
enhancement ratios ($\Delta CO/\Delta CO_2$) across flight groups and along vertical profiles. These ratios
represent the change of CO abundance per unit change in $CO_2$ relative to their corresponding
background values (i.e., enhancement or excess). Here, enhancement ratios refer to the slopes
derived from a bivariate linear regression of CO and $CO_2$ data points rather than the estimates of
the ratio of enhancements based on *a priori* knowledge of their background (e.g., Yokelson et al.,
2013; Hedelius et al., 2018). The results in our model evaluation against KORUS-AQ
measurements indicate that at near surface and near polluted conditions, both CO and $CO_2$ are
underestimated suggesting a possible underestimation of common local processes, while aloft, $CO_2$
is slightly overestimated, and CO is underestimated suggesting a more dominant "background"
influence. Here, we will assess if variations in $R_{CO,CO_2}$ and $\Delta CO/\Delta CO_2$ also reflect this finding.
We have broken down the statistics in Table 2, with regards to modeled and observed correlation
between CO and $CO_2$ and their associated error correlations, into 6 (1-km) vertical layers for each
flight group. We also derived the corresponding vertical profile of $\Delta CO/\Delta CO_2$ using two
regression approaches: 1) ordinary least squares (OLS) regression approach with $CO_2$ as our
predictor since it is more stable than CO, and 2) reduced major axis regression (RMA) at 95%
confidence to account for errors in both CO and $CO_2$. The enhancement ratios in Table 2
correspond to regression slopes using RMA. We will refer to CAM-chem simulations with CT3h
fluxes as the model equivalents for all our analyses.





### 4.1 Correlation and error correlation

Figure 6 shows the vertical profiles of the $CO:CO_2$ statistics for each flight group during the campaign. Although we only plotted statistically significant correlation and error correlation, statistics using less than 30 data points are not considered in this analysis. It is important to note here that these statistics are only indicative of covariations in CO and $CO_2$. We focus our analysis on the relative differences between observed and modeled statistics. This only serves as another piece of information on the variability in CO and $CO_2$ relationship in the region.

Below 2 km, the modeled $CO:CO_2$ correlation ($R_{CO,CO_2}^{mod}$) is systematically lower than observed ($R_{CO,CO_2}^{obs}$) except at 1 km in Seoul-Jeju and Seoul-Busan. The average $R_{CO,CO_2}^{obs}$ values across flight groups is 0.67±0.02 whereas the average $R_{CO,CO_2}^{obs}$ is 0.47±0.16. Aloft, it is the opposite (i.e., modeled correlation is higher than observed) except in Taehwa, where they appear to be diverging along the upper layer of the vertical profile. The average $R_{CO,CO_2}^{obs}$ value across flight groups in these vertical levels is 0.47±0.22, whereas the average $R_{CO,CO_2}^{obs}$ is 0.55±0.32. This pattern of lower modeled correlation at the surface but higher aloft is clearly seen in West Sea, where we see the highest $R_{CO,CO_2}^{obs}$ (0.95) against the lowest $R_{CO,CO_2}^{mod}$ (0.11) among flight groups. The low $R_{CO,CO_2}^{mod}$ relative to $R_{CO,CO_2}^{obs}$ at the surface supports previous discussion that the model does not capture the observed variability in both CO and $CO_2$ data. Near the surface, a high $R_{CO,CO_2}$ in both model and observations can be associated with well-correlated sources and sinks since CO, $CO_2$, and ff$CO_2$ share the same model transport representation. A low $R_{CO,CO_2}^{mod}$ but high $R_{CO,CO_2}^{obs}$ on the other hand, can be associated with lack of variability in the model. Similar underestimation of boundary layer $R_{CO,CO_2}^{mod}$ (albeit notably smaller) can be found in Seoul (0.57 vs 0.79) and Taehwa (0.41 vs 0.61). Coarser spatiotemporal representation of associated sources and sinks and boundary layer processes can influence these values. In Tang et al. (2018), for example, we find that the 9-km resolution forecast/analysis of CAMS with 137 vertical levels (FC9s) led to significantly closer correlation to $R_{CO,CO_2}^{obs}$ than its free running 16-km resolution (FC16s), except over West Sea where both FC16s and FC9s, like in CAM-chem, failed to capture the high $R_{CO,CO_2}^{obs}$.

On the other hand, above 2-3 km, $R_{CO,CO_2}^{mod}$ is higher than $R_{CO,CO_2}^{obs}$ indicating that the modeled air masses are more influenced by relatively less-aged plumes transported into the region. As we will discussed in later section, the influence of emissions to CO and $CO_2$ over Korea are significantly limited to the boundary layer and hence, the vertical profiles of these correlations exhibit a strong contrast on local and regional influences in the sampling region. During TRACE-P (2001), the $R_{CO,CO_2}^{obs}$ coefficients reported by Palmer et al. (2006) using GEOS-Chem is mostly greater than 0.7 varying only within 5-10%. They observed lower $R_{CO,CO_2}^{obs}$ aloft which they attribute to a larger influence of aged air masses from Asia. While noting that the flights during TRACE-P is farther downwind (and has a larger coverage) than KORUS-AQ flights, we see a similar pattern (albeit lower in magnitude) to $R_{CO,CO_2}^{obs}$ during KORUS-AQ. The lower magnitudes are due to higher background values (and more variable) in KORUS-AQ than TRACE-P, following the same reasoning by Palmer et al. (2006) for relatively polluted TRACE-P samples located >30 degrees north. These differences highlight the importance of vertical information in effectively differentiating local and regional influences (and associated errors in transport versus emissions),





especially within an inverse modeling framework (e.g., Stephens et al., 2007, Schuh et al., 2019,
Arellano et al., 2006, Jiang et al., 2015).
Vertical profiles of the error correlation between CO and $CO_2$ ( $errR_{CO,CO_2}$) provide a
complementary perspective in examining biases in the model and in quantifying model-data error
covariances used in inverse modeling algorithms. A high $errR_{CO,CO_2}$ corresponds to a higher
correlation between the errors in CO and $CO_2$, while a low $errR_{CO,CO_2}$ indicates the presence of
model misrepresentation of processes on either $CO_2$ or CO that are not related to the other (i.e.,
different sources and sinks). Although the overall $errR_{CO,CO_2}$ values in CAM-chem is smaller than
we previously reported for CAMS, the $errR_{CO,CO_2}$ values in CAMS are also lower compared to
$R^{obs}_{CO,CO_2}$. We note that $errR_{CO,CO_2}$ values in both CAM-Chem (0.57) and CAMS (0.82) are highest
over West Sea among flight groups, regardless of resolution in the case of CAMS. Furthermore,
over West Sea, the $errR_{CO,CO_2}$ in CAM-chem near the surface (0.5 km) lies in the middle of its
$R^{obs}_{CO,CO_2}$ and $R^{mod}_{CO,CO_2}$. Values of $errR_{CO,CO_2}$ that are closer towards $R^{obs}_{CO,CO_2}$ are interpreted to reflect
errors in CO and $CO_2$ processes that are related (i.e., common sources and sinks). This indicates
that East Asian sources are clearly the dominant influence on $errR_{CO,CO_2}$ for these samples; more
than their associated sinks during transport, since over Yellow Sea, CO and $CO_2$ do not share a
common major sink. Differences between modeled and observed correlation can be associated
with coarser representation of related processes. On the other hand, over Seoul, CAM-chem
$errR_{CO,CO_2}$ (0.36) is smaller than CAMS (0.64). The value in CAMS is the second highest among
flight groups, while the value in CAM-chem is the lowest. The $errR_{CO,CO_2}$ over Seoul (0.35) in
CAM-chem near the surface (0.5 km) is lower than both $R^{obs}_{CO,CO_2}$ and $R^{mod}_{CO,CO_2}$. Model
misrepresentation of unrelated processes may also be influencing these values (e.g., secondary CO,
non-ff$CO_2$). We note that the pattern in $errR_{CO,CO_2}$ along the overall vertical profile is consistent
(albeit lower in magnitude) with $R^{obs}_{CO,CO_2}$ (except at 4-5 km where it follows $R^{mod}_{CO,CO_2}$). Patterns in
other flight groups cannot be compared due to incomplete statistically significant data points.
**4.2    Enhancement ratios**
Vertical profiles of modeled and observed $\Delta CO/\Delta CO_2$ are also shown in Figure 6. Like in previous
section, please note that data points in the profile which are not statistically significant in
correlation and having less than 30 points are not considered in this analysis to avoid
misinterpretation of results. Also, estimates of slopes derived from both OLS and RMA regression
are plotted in Figure 6 to show the difference due to the choice of regression algorithm. Although
both slope estimates follow the same pattern along the vertical profile, the slopes from OLS is
systematically lower by 50%. The OLS algorithm is useful in understanding patterns rather than
in comparing magnitudes with other studies.  In OLS, $\Delta CO/\Delta CO_2$ can be expressed as the product
of $R_{CO,CO_2}$ and the ratio of the respective standard deviations ($\sigma_{CO}/\sigma_{CO_2}$). As such, the difference
between OLS $\Delta CO/\Delta CO_2$ and $R_{CO,CO_2}$ profiles correspond to $\sigma_{CO}/\sigma_{CO_2}$, for which such quantity
can be better represented in RMA regression.
Overall, the observed and modeled RMA $\Delta CO/\Delta CO_2$ across all altitudes are very similar, with
values of 13.30±0.21 ppb/ppm (~1.3%) and 13.80±0.23 ppb/ppm (~1.4%), respectively (see scatter
plot in Figure S6). Higher values of $\Delta CO/\Delta CO_2$ correspond to air masses that are characterized (in
a bulk average sense) as less efficient (i.e., high CO is associated with low temperature and less



efficient combustion). However, it should be noted that as Halliday et al. (2019) pointed out, these
values when viewed as bulk efficiency, are limited only to bulk emission ratio interpretation since
these regressions are subject to transport and mixing processes as well. Values that are derived
from short-term covariations of CO and $CO_2$ are more useful for air mass characterization since
these ratios are non-stationary in both space and time. Variations across flight groups – here
representing non-stationarity in horizontal space --- (Seoul: 9.1, West Sea: 28.2, Taehwa: 15.3,
Seoul-Busan: 15.9, and Seoul-Jeju: 10.4 ppb/ppm) are also captured well by the model (Seoul:
12.6, West Sea: 33.7, Taehwa: 16.6, Seoul-Busan: 10.7, and Seoul-Jeju: 11.5. ppb/ppm). The
overall observed value of 13.30 ppb/ppm reflects the influence of relatively more efficient air
masses from Korea (flight groups other than West Sea) and less efficient air masses from China
(West Sea flight group)(see Figure S6 as well). The variability across flight groups within Korea
(Seoul and Seoul-Jeju versus Seoul-Busan and Taehwa) is likely due to a mixture of source
influences across these locations (i.e., biogenic CO sources and biospheric influence on $CO_2$ over
Taehwa and Seoul-Busan). These model values are comparable (albeit closer to observed values)
to values from the best simulation of CAMS (FC9s) in Tang et al. (2018).
Similar to the correlation profiles, the modeled $\Delta CO/\Delta CO_2$ show larger differences against
observed $\Delta CO/\Delta CO_2$ along the vertical profile. The observed values in All flights are 5.9, 11.8,
11.2, 10.8, 2.8, and 6.7 ppb/ppm for layers from 0.5 to 5.5 km at 1 km interval, respectively. This
variability with height was also pointed out by Halliday et al. (2019). Higher values can be seen
especially at 1.5-3.5 km. The differences between modeled and observed $\Delta CO/\Delta CO_2$ are also more
pronounced above 3 km (see All flights). It is interesting to note as well that the modeled values
at the surface from RMA regression in West Sea (21.6) and Seoul (11.2) are similar to observed
values (23 ppb/ppm for West Sea and 8 ppb/ppm for Seoul). Again, this suggests that the
differences in $R_{CO,CO_2}$ found in West Sea are mostly due to misrepresentation of related processes
rather than unrelated processes. This is reflected in the lower slope from OLS that matches with
low $R_{CO,CO_2}$. The slopes from RMA are associated more to $\sigma_{CO}/\sigma_{CO_2}$ which indicate more of a
signature from sources and sinks than transport-related processes in $R_{CO,CO_2}$. This can be shown in
the overestimation of modeled $\Delta CO/\Delta CO_2$ at 2-3 km by 40 ppb/ppm in West Sea and 29 ppb/ppm
in All flights. This suggest an overestimation of emission ratio from regional sources (i.e., East
Asia). This is also reflected in the larger overestimation in CO (67 ppb and 80 ppb) at this level
over Seoul and West Sea (8 ppb in All flight) and only slight overestimation in $CO_2$ (0.4 to 1.2
ppm) consistent with our earlier discussion on biases.
Relative to other $\Delta CO/\Delta CO_2$ values reported in this region, the observed $\Delta CO/\Delta CO_2$ during
KORUS-AQ shows a similar bulk combustion efficiency contrast between South Korea and China
(i.e., 9 ppb/ppm in Seoul against 28 ppb/ppm in West Sea). During this campaign, the observed
$\Delta CO/\Delta CO_2$ from the ARIAs campaign over China (Benish et al., 2020) is also larger than 20
ppb/ppm. Fifteen years prior to KORUS-AQ and ARIAs, $\Delta CO/\Delta CO_2$ from northern China during
TRACE-P in 2001 was observed to be largely higher (50-100 ppb/ppm) than over Japan (~12-17
ppb/ppm) (Suntharalingam et al., 2004). A similar contrast (albeit weaker than TRACE-P) was
also reported by Turnbull et al. (2011) in terms of CO:CO2ff ratios over Shangdianzi, China (~47
ppb/ppm) and South Korea (13 ppb/ppm) during winter 2009/2010. This is consistent with the
downward change in $\Delta CO/\Delta CO_2$ near Beijing from 34-42 ppb/ppm in 2005-2007 to 22 ppb/ppm
in 2008 (Wang et al., 2010) and derived $\Delta CO/\Delta CO_2$ from GOSAT/ACOS and MOPITT retrievals
over Seoul (~7-9 ± 0.5 ppb/ppm) and Beijing (~43 ±6 ppb/ppm) in 2010 (Silva et al., 2013). As





we have previously noted, we expect that as combustion activities become more efficient in China,
this contrast will decrease in recent years. Unfortunately, there are very limited measurements
(even in TCCON AMY, Goo et al., 2017, and NOAA/KMA-GAW/AMY sites) that we can use to
investigate these possible changes. The recent study by Lee et al. (2020) reports similar values
($\Delta CO/\Delta ffCO_2$) derived from NOAA/KMA-GAW/AMY site for air masses coming from the Asian
continent (29-36 ppb/ppm) and Korea (8±2 ppb/ppm) during May 2014 to August 2016. Another
recent study by Xia et al. (2020) also reports a mean $\Delta CO/\Delta CO_2$ of 21.6 ppb/ppm over Jingdezhen
(JDZ) site in central China during the winter months of 2018 to 2019. Together with $ffCO_2$ data
(section 3.2), there appears to be a decrease in this contrast relative to TRACE-P, possibly due to
improved efficiency in both China and Korea energy and road transportation sectors. Activities,
like biofuel and biomass burning, which have lower combustion efficiency, may still influence the
higher ratios in China (e.g., Chen et al. 2017). However, this possibility needs to be verified with
correlative measurements having sufficient spatiotemporal coverage of the region. As has been
suggested in past studies (e.g., Turnbull et al. 2006; Vardag et al. 2015; Super, 2018; Halliday et
al. 2019), these comparisons across flight groups, sampling locations, altitude, and time highlight
the importance of understanding and properly accounting for the spatiotemporal variability of
$\Delta CO/\Delta CO_2$ when estimating $ffCO_2$ emissions since differences in $\Delta CO/\Delta CO_2$ have confounding
factors and cannot be directly attributed to discrepancies in emissions unless investigated
appropriately.

### 651    4.3    Local and regional contributions

We use the tagged ffCO and $ffCO_2$ simulations to further elucidate the contributions of local and
regional influences on inferred relationships of CO, $CO_2$, and $ffCO_2$ during the campaign. We
show in Figure 7 the spatial distribution of modeled $CO_2$, CO, and $ffCO_2$ including the associated
distribution of $ffCO_2$ tags at three representative vertical levels (model surface, 800 hPa or ~2 km,
500 hPa or ~5 km above sea level). We also show in Figure S7 a zoom-in version with a side-by-
side comparison of $CO_2$ and CO and their associated tags at the surface and also across the mean
vertical profile. The moderately strong relationship between surface $CO_2$ and $ffCO_2$ (0.82), which
is evident over areas of fossil fuel and biofuel combustion, is also found in the relationship between
surface $CO_2$ and CO (0.84). However, there is a high $CO_2$ signature over Seoul and EA-S that is
not very apparent in CO, as has been noted in our OCO-2 and MOPITT qualitative assessment.
High $CO_2$ signatures in the model are associated with mostly $ffCO_2$ (EA-M, EA-N) and fire (EA-
S) emissions. Unlike $CO_2$ and $ffCO_2$, the similarity between $CO_2$ and CO is degraded at higher
altitudes (0.66-0.68) due to regional and background influences in CO since $ffCO_2$ aloft is not
affected by its surface sinks. Note that East Asian and ROW $ffCO_2$ also account for the majority
of $ffCO_2$ at these levels, clearly indicating regional influences on the air aloft during the campaign.
This is evident as well from the associated flight curtains of these tags relative to modeled $CO_2$
shown in Figure S8 (All group) and Figure S9 (West Sea group).
To quantify the contribution of local and regional influences of $ffCO_2$ to observed $\Delta CO/\Delta CO_2$, we
decompose the modeled $\Delta CO/\Delta CO_2$ into four basis functions. The observed $CO_2$ can be
represented as the sum of $ffCO_2$ abundance (or response functions) from Korea and Japan
(hereinafter Kor+Jap), East Asia, and ROW $ffCO_2$ sources (or basis functions), along with other
contributions (non-$ffCO_2$ and background), termed here as "Background+non-$ffCO_2$" (see
Appendix A). We can then regress each response function to the observed CO and $CO_2$ following





the approach used by Cheng et al. (2018) in decomposing the contributions of tagged $CH_2O$ to
$\Delta O_3 / \Delta CH_2O$. That is,
$$\left. \Delta CO \middle/ \Delta CO_2 \right|_{obs} \approx \sum_{i=1}^{i=4} \left( cov(basis_i, CO^{obs}) \middle/ var(CO_2{}^{obs}) \right) \qquad \qquad Eq. (1)$$
where $basis_i$ corresponds to $ffCO_2{}^{opt}_{Kor+Jap}$, $ffCO_2{}^{opt}_{East\ Asia}$, $ffCO_2{}^{opt}_{ROW}$ or $bg + nonffCO_2$.
This "Background+non-ffCO$_2$" is calculated as the difference between $CO_2{}^{obs}$ and the sum of
$ffCO_2{}^{opt}$. To ensure that ffCO$_2$ closely matches with derived ffCO$_2$ measurements (section 3),
the tagged ffCO$_2$ abundances were optimized through a Bayesian synthesis inversion, which we
will describe in the next section. An alternative to tagged simulations is the backward trajectory
analysis using FLEXPART (Stohl et al. 2009), STILT (Lin et al., 2003), or HYSPLIT (Draxler et
al., 1997). This has been used in past studies for a similar analysis (Turnbull et al., 2011; Vardag
et al. 2015; Xia et al., 2020).
Here, we regress each response function with the CO and CO$_2$ data for each flight group and for
each group of vertical bins (<1.5 km, 1.5-3.0 km, and >3.0 km), in order to examine the ffCO$_2$
contributions to the enhancement ratios discussed in previous section. These contributions are
shown in Figure 8, together with the slope estimates from observations of CO and CO$_2$ using OLS
regression. It is clear from this result that the influence of "Background+non-ffCO$_2$" dominates
across the vertical levels, even near the surface and polluted conditions in Seoul. This can be seen
across all flight groups, where the median contributions for each bin are ~74% for <1.5km, ~47%
for 1.5-3 km, and ~81% for >3 km. We also find that ffCO$_2$ contributions in the West Sea flight
group at 1.5-3.0 km and >3 km bins are dominated by ffCO$_2$ from East Asia (~67% for 1.5-3.0
km, ~131% for >3km), with the "Background+non-ffCO$_2$" contributing 90% at the surface and
negatively (-52%) on the air aloft. The dominance of "Background+non-ffCO$_2$" suggests that the
low $R^{mod}_{CO,CO_2}$ relative to $R^{obs}_{CO,CO_2}$, yet consistent $\Delta CO / \Delta CO_2$ at the surface of the West Sea flight
group, can be attributed to possible inability of the model to represent spatiotemporally finer
variations in both non-ffCO$_2$ and background transport from East Asia, rather than inconsistency
in ffCO$_2$ emission ratio for this region. However, it is clear that the air just above 2 km is
characterized to be a low efficient airmass (high $\Delta CO / \Delta CO_2$), having higher $R^{mod}_{CO,CO_2}$ than $R^{obs}_{CO,CO_2}$
yet consistent $errR_{CO,CO_2}$ $R^{obs}_{CO,CO_2}$ and very high East Asian influence. These conditions clearly
indicate an overestimation of emission ratio in East Asia. While we are aware that ffCO$_2$ and CO
emissions used in this study are taken from different emission inventories which may have caused
this overestimation, this highlights a regional inconsistency between inventories.
The contribution of ffCO$_2$ from Kor+Jap is relatively small, even at the surface (<1.5km) in Seoul
(29%), Seoul-Jeju (20%), Taehwa (15%), and Seoul-Busan (13%). Its contribution can also be
seen at 1.5-3 km in Seoul-Busan (27%) and Taehwa (15%). Above 3 km, this influence is very
minimal, even in Seoul-Busan (0.8%) and Taehwa (2%). In contrast, the contribution of ffCO$_2$
from East Asia is relatively high, even at the surface in Korea (Seoul: 12%, Seoul-Busan: 21%).
Above 1.5 km, the East Asian influence over these flight groups are significant (35% for 1.5-
3.0km, 20% for >3km) relative to Kor+Jap. These results strongly suggest that while regional
influence can be inferred, it is critical to understand the vertical structure of these response
functions and recognize the large influence of regional emissions and background on the local
environment. The long-range transport of pollution into the region is known to be present. Simpson


et al. (2020) also found a larger contribution of CO from long-range transport in the Seoul
Metropolitan Area than CO from combustion over Seoul. The signal-to-noise for ffCO$_2$ abundance
is very low compared to the biospheric fluxes, model transport errors, and source estimation
methods (Schuh et al., 2019; Crowell et al., 2019). Accurately estimating ffCO$_2$ emissions at local-
to-regional scales requires sufficient data coverage and precision, especially within the boundary
layer. The statistics that we have presented also points to reducing representativeness and
aggregation errors through the use of higher resolution models, which are expected to be able to
capture the local scale variations. Although CAM-chem at current resolution (0.9 deg x 1 deg) is
able to represent the regional-scale transport, the presence of confounding factors in the boundary
layer limits our ability to improve the signal-to-noise and our ability to exploit all datasets given
that associated errors are sensitive to sampling characteristics. These have been highlighted in
current studies of potential ffCO$_2$ network (Wang et al., 2017; 2018). Furthermore, exploiting the
finer spatiotemporal scale signatures of ffCO$_2$ on CO$_2$ data, which can serve as valuable
observational constraint (e.g., Shiga et al., 2014; Liu et al., 2017), cannot be exploited at coarser
resolution. Variations across the vertical has implications as well on inversions using columnar
data from satellite retrievals of XCO$_2$.

## 5. Joint CO:CO$_2$ inversions

We saw from the results discussed above that there are spatial variations in CO$_2$ (and CO)
attributable to East Asian underestimation (overestimation) and overestimation (underestimation)
of "background" conditions. It is more complicated, however, to attribute a Korean
underestimation (overestimation) as competing local processes are present. As we have
demonstrated, using information on CO$_2$ and CO relationship provides more context to this
problem in lieu of ffCO$_2$ data. To demonstrate the potential of CO data in refining estimates of
regional ffCO$_2$ emissions, we conducted three sets of Bayesian synthesis inversions following what
we learned from our model evaluation and analysis of CO, CO$_2$ and ffCO$_2$ and their associated
relationships (section 3 and 4). We conducted two single-species experiments: 1) using ffCO$_2$ data,
and 2) using CO$_2$ data, as well as, one joint inversion using both CO$_2$ and CO data. These inversion
experiments are designed simply to quantify the broader role of CO in refining regional scale ffCO$_2$
signatures, which is expected to complement the current yet relatively sparse ffCO$_2$ observing
system and the national networks proposed (e.g., Basu et al., 2016; Wang et al., 2018). We revisit
the Bayesian synthesis inversion algorithm used in one of the first studies of joint regional CO:CO$_2$
inversion with aircraft data from TRACE-P and GEOS-Chem by Palmer et al. (2006). A recent
study by Boschetti et al. (2018) used a similar method using IAGOS CO, CO$_2$ and CH$_4$ data
(Petzold et al., 2015) and STILT to conduct OSSEs for global multi-species inversions. This
approach has also been used in the past for single atmospheric constituent inversions (e.g., Enting,
2002; Baker et al., 2006; Wang et al., 2018). This approach begins with the assumption of a linear
relationship between observation and model, i.e.,
$$\mathbf{y} = \mathbf{Kx} + \mathbf{e}_y \, , \qquad\qquad Eq. (2)$$
where $\mathbf{y}$ is a vector of observations (in our case: ffCO$_2$, CO, and/or CO$_2$), $\mathbf{x}$ is a vector of time
averaged source strengths (or basis functions, which in our case is mainly ffCO$_2$ Kor+Jap, ffCO$_2$
East Asia and ffCO$_2$ ROW). $\mathbf{K}$ is a matrix of contribution (or response functions) calculated from
our tagged simulations, and $\mathbf{e}_y$ is a vector of errors associated to both $\mathbf{K}$ and $\mathbf{y}$. Assuming Gaussian
unbiased error statistics on both $\mathbf{e}_y$ and the error $\mathbf{e}_x$ on the *a priori* source strengths having average





values represented as a vector $\mathbf{x_a}$, the solution to this Bayesian problem is the maximum a
posteriori (MAP) solution:
$$\hat{\mathbf{x}} = (\mathbf{K}^T\mathbf{S_e}^{-1}\mathbf{K} + \mathbf{S_a}^{-1})^{-1}(\mathbf{K}^T\mathbf{S_e}^{-1}\mathbf{y} + \mathbf{S_a}^{-1}\mathbf{x_a}) , \quad \hat{\mathbf{S}} = (\mathbf{K}^T\mathbf{S_e}^{-1}\mathbf{K} + \mathbf{S_a}^{-1})^{-1}$$  *Eq. (3)*
where $\hat{\mathbf{x}}$ and $\hat{\mathbf{S}}$ are *posteriori* mean and error covariance estimates, respectively. $\mathbf{S_e}$ and $\mathbf{S_a}$ are the
expected observation $\langle \mathbf{e_y}\mathbf{e_y}^T \rangle$ and *a priori* source $\langle \mathbf{e_x}\mathbf{e_x}^T \rangle$ error covariance matrices, respectively.
Superscript $^T$ denotes transpose, $^{-1}$ the inverse of a matrix and $\langle \ \ \rangle$ is an expectation operator. These
notations follow Rodgers (2000). Note that this approach suffers from wrong
assumptions/misspecification of the error covariances, especially $\mathbf{S_e}$, which includes not only
instrument/retrieval noise but more importantly errors in $\mathbf{K}$ when translating emissions to
abundance (i.e., transport and vertical mixing errors in the tagged simulations). Here, we take a
similar approach by Palmer et al. (2006) and Wang et al. (2017), where we estimate $\mathbf{S_e}$ from the
error statistics we obtained in previous section. That is, $\mathbf{S_e}^{ffCO_2}$, $\mathbf{S_e}^{CO_2}$ and $\mathbf{S_e}^{CO}$ are assumed to be
diagonal matrices with the elements corresponding to $(\mathbf{e_y})^2 = (1\ ppm)^2$, $(\mathbf{e_y})^2 = (0.01\mathbf{y}\ ppm)^2$ and
$(\mathbf{e_y})^2 = (0.2\mathbf{y}\ ppb)^2$, respectively. Note that the error variances in $CO_2$ and CO are relative quantities
represented as fractions of the data magnitude. We also inflate these fractions to account for
representativeness errors. In the case of joint CO:$CO_2$ inversion, we augment the observation
vector such that $\mathbf{y} = [\mathbf{y}^{CO_2}, \ \mathbf{y}^{CO}]^T$. We also use the error correlation between CO and $CO_2$
discussed in previous section. That is, $\mathbf{S_e}$ can be expressed as:
$$\mathbf{S_e} = \begin{bmatrix} \mathbf{I}_{n_y}(e_y^{CO_2})^2 & \mathbf{I}_{n_y}errR_{CO,CO_2} \\ \mathbf{I}_{n_y}errR_{CO,CO_2} & \mathbf{I}_{n_y}(e_y^{CO})^2 \end{bmatrix}$$  *Eq. (4)*
where $\mathbf{I}_{n_y}$ is an identity matrix with $n_y$ diagonal elements corresponding to the number of data
points for each species. Here, we use a much lower $errR_{CO,CO_2}$ of 0.33. Notice that in Palmer et
al. (2006), they used $R_{CO,CO_2}$ (0.7) on $\mathbf{S_e}$, which is much higher than the model-dependent
$errR_{CO,CO_2}$ from this study. A similar error correlation of 0.7 was also used by Boschetti et al.
(2018). While we recognize that from a purist perspective, $\mathbf{S_e}$ should only account errors in the
data, we also need to account for model errors (in observation space) as the assumption of perfect
$\mathbf{K}$ is obviously not valid. We use the more conservative $errR_{CO,CO_2}$ to represent the correlation
component of $\mathbf{S_e}$ assuming that these model errors are more reflected in correlation than the
variance structure of $\mathbf{S_e}$. However, we still use the errors on the data to represent the error variance
component of $\mathbf{S_e}$ but with added inflation to account for representativeness errors (which is also
model-dependent). Albeit clearly simplified, this is along the same line as the more rigorous
representation of these errors discussed in Wang et al. (2017, 2018) and Basu et al. (2016). We
also filter the data with points having the residual (model-obs) variance that is a factor of 1.25 (for
ffCO$_2$ data) or 2.0 (for CO$_2$ and CO data) greater than the overall residual standard deviation. More
importantly, we only use data below 3 km for localization purposes (see previous section). The
effective number of data points for each observation vector that are used in a particular inversion
are as follows: $n_y^{fCO_2}$=41, $n_y^{CO_2}$=4,716, and $n_y^{CO}$=4,716. Notice that exactly the same set of CO$_2$
data points in CO$_2$ inversion is used for the joint CO:CO$_2$ inversion to facilitate comparison
between inversions. Our emphasis for these inversions is to show the role of CO in refining our





estimates of ffCO$_2$ emissions rather than accurately estimating biospheric sources and sinks. For
the same reason that we use *a posteriori* CO$_2$ fluxes rather than the *a priori* CAM-chem fluxes.
For single-species inversion using ffCO$_2$ data, we added another basis that we call 'ffCO$_2$ Offset'.
This is a constant term (1 ppm) that is intended to account for a potential bias in ffCO$_2$ due to our
assumption of ffCO$_2$ initial condition. We replace the basis "ffCO$_2$ Offset" for the single-species
inversion using CO$_2$ data with the residual between modeled CO$_2$ and modeled ffCO$_2$ and call it
"Background+non-ffCO$_2$" as noted in section 4.3. This represents the larger non-ffCO$_2$ component
of CO$_2$ (see Eq. A.1). Both single-species inversions will have m=4 basis functions that will be
optimized using Eq. 3.  For joint inversion, there will be m=8 basis functions corresponding to
CO$_2$ and CO basis (i.e., $\mathbf{x_a} = \begin{bmatrix} x_a^{CO_2} & x_a^{CO} \end{bmatrix}^T$). The 4x4 $\mathbf{S_a}$ matrix for single species ffCO$_2$ inversion
is assumed to be diagonal with $\mathbf{e_x} = \boldsymbol{d} \circ \mathbf{x_a}$ and $\boldsymbol{d} = [0.3, 0.3, 0.1, 0.5]^T$ to account for
heteroskedasticity in these errors. We assumed that error in ffCO$_2$ ROW is the smallest while the
"ffCO$_2$ Offset" is largest. However, as we mentioned before, the $^{14}$CO$_2$-derived ffCO$_2$ is
representative of the regional ffCO$_2$ (not global) and specific to the assumptions of $\Delta^{14}$CO$_2$. We
have seen from section 4.3 as well that ffCO$_2$ ROW has negligible contributions to $\Delta CO/\Delta CO_2$ in
the region. We expect that the errors in ffCO$_2$ ROW and "ffCO$_2$ Offset" to be largely correlated.
Accordingly, the 8x8 $\mathbf{S_a}$ matrix for the joint CO:CO$_2$ inversion is constructed as follows:
$$\mathbf{S_a} = \mathbf{s} \cdot \mathbf{C_a} \cdot \mathbf{s}, \text{ where } \mathbf{s} = \begin{bmatrix} I_4\left(e_x^{CO_2}\right)^2 & 0 \\ 0 & I_4(e_x^{CO})^2 \end{bmatrix} \text{ and } \mathbf{C_a} = \begin{bmatrix} I_4 & I_4 c \\ I_4 c & I_4 \end{bmatrix} \qquad Eq. (5)$$
We assumed no correlation across basis functions within a particular species. However, the source
error correlation across species is specified as $\mathbf{c} = [-0.5, -0.5, -0.1, 0.0]^T$. We also assumed that
the source error correlation across species is higher near the source region (i.e., East Asia and
Kor+Jap) and smaller to negligible for the more "diffused" sources from ROW and
"Background+non-ffCO$_2$". At the source, CO is mostly negatively correlated with CO$_2$ (i.e., higher
combustion efficiency is associated with low CO). It should be noted that while this vector is
critical in transferring information from CO (or CO$_2$) data to the other species (Palmer et al., 2006,
Boschetti et al. 2018), there is little information on quantifying this correlation. In fact, it is very
difficult to accurately specify the elements of $\mathbf{C_a}$ since these statistics cannot be derived from
measurements. There are only few direct measurements of CO$_2$ fluxes (and CO emissions) to
quantify their associated errors. One way to estimate $\mathbf{C_a}$ is to have an ensemble of CO and CO$_2$
sources, where we can compute its statistics following a similar approach by Wang et al. (2018).
For this study, however, we follow a simpler approach using similar critical values of these
correlations suggested in Palmer et al. (2006). This is more conservative than the correlation used
by Boschetti et al. (2018) of 0.7. We note that in our setup, *a posteriori* estimates are not that
sensitive to the correlation values in $\mathbf{S_a}$ than in $\mathbf{S_e}$. We also specify the error variances while
accounting for heteroskedasticity as:  $e_x^{CO_2} = \boldsymbol{d_{CO_2}} \circ \mathbf{x_a^{CO_2}}$ where $\boldsymbol{d_{CO_2}} = [0.3, 0.3, 0.1, 0.05]^T$
and $e_x^{CO} = \boldsymbol{d_{CO}} \circ \mathbf{x_a^{CO}}$ where $\boldsymbol{d_{CO}} = [0.5, 0.5, 0.1, 0.05]^T$. These error variances are typically
prescribed to be larger than reported 1-*sigma* uncertainties in order to include potential errors that
are unaccounted for. We assumed that errors in ffCO emissions are larger in East Asia, and
Kor+Jap than in ROW while the "Background+non-ffCO" is smallest based on their associated
variability.





### 5.1 Inversion results

We present in Figure 9 the results of the three sets of inversions. We show the change in *a posteriori* estimate relative to *a priori* (represented here as scaling factors) of ffCO₂ basis including "ffCO₂ Offset" or "Background+non-ffCO₂" (depending on the dataset used in the inversion). The error bars correspond to the square root of the diagonal elements of $\hat{\mathbf{S}}$ for *a posteriori* or $\mathbf{S_a}$ for *a priori* estimates. The error for *a priori* "Background+non-ffCO₂" is not shown. For ffCO₂ inversion, we find that ffCO₂ East Asia and ffCO₂ Kor+Jap need to be increased by ~27% ±9% and ~10% ±3%, respectively. At the same time, ffCO₂ ROW needs to be slightly decreased (albeit with higher uncertainty) by 14% ±9%. This results to a reduction in bias (model-obs) against ffCO₂ derived measurements (including "ffCO₂ Offset") from -1 ppm to -0.01 ppm. The error reduction in ffCO₂ estimates (1- $\hat{e}_x/e_x$), where $\hat{e}_x$ is the *a posteriori* error, is largest in ffCO₂ Kor+Jap (91%) followed by ffCO₂ East Asia (71%), "ffCO₂ Offset" (62%), and ffCO₂ ROW (8%), suggesting that East Asia and Kor+Jap are reasonably resolved by the measurements. Again, it is important to note that we do not expect ¹⁴CO₂-derived ffCO₂ measurements to resolve ffCO₂ ROW. The error reductions in East Asia and Kor+Jap are comparable to the uncertainty reduction (UR) values reported in Wang et al. (2018) for OSSEs using a potential ffCO₂ network in Europe. The increases in East Asia and Kor+Jap are also expected based on our evaluation of modeled CO₂ and ffCO₂ (section 3) and our analysis of CO and CO₂ relationships (section 4) of apparent underestimation of CO₂, and ffCO₂ below 3 km. Although such increase is reasonable and within range of the uncertainties in regional ffCO₂ emissions (Andres et al., 2012), the equivalent reduction of the bias in terms of CO₂ abundance remains small, even with the contribution of "ffCO₂ Offset". This is consistent with the relatively low contribution of ffCO₂ from these source regions discussed in section 4.3.

We find reasonable consistency in scaling factors that are within the range of their associated uncertainties when CO₂ and CO across the campaign are used instead of ffCO₂ data. In particular, emissions of ffCO₂ from East Asia and Kor+Jap need to be increased by ~27% ±24% and (9% ±17%). However, the scaling factor for ffCO₂ from ROW only suggests a smaller decrease (6%±10%) in ffCO₂ emissions compared to ffCO₂ inversion. The "Background+non-ffCO₂" appears to only have a very small decrease (0.7% ±0.3%). Reduction in the error estimates are lower (although still significant) in East Asia (20%) and Kor+Jap (42%). On the other hand, there is very little error reduction in ROW (0.4%) but higher error reduction in "Background+non-ffCO₂" (94%) indicating that the estimate of ffCO₂ from ROW is not resolved using either CO, CO₂ or ffCO₂ measurements. This is expected as the source error correlation for this basis function is smaller and that the contribution of ffCO₂ ROW is already very small to begin with. On the other hand, the error reduction in "Background+non-ffCO₂" is mostly constrained by CO₂ data given that we assumed zero source error correlation across species. However, unlike the joint inversion, we find larger differences in ffCO₂ mean estimates when CO₂ measurements across the campaign are used. Our results show a decrease in both ffCO₂ East Asia (5% ±27%) and Kor+Jap (6% ±19%) and practically no changes in ffCO₂ ROW (0% ±10%) and "Background+non-ffCO₂" (0% ±0.3%). The error reduction is slightly smaller than the reduction from joint inversion for East Asia (9%) and Kor+Jap (38%), while similar error reduction can be observed for ROW (0.1%) and "Background CO₂" (94%), again suggesting that ffCO₂ ROW is not resolved neither by CO₂ nor CO measurements as well.



## 6. Discussion and general implications


These results imply that inversion using CO and $CO_2$ data is able to match the regional $ffCO_2$
emission estimates for East Asia and Kor+Jap from $ffCO_2$ inversion, whereas using $CO_2$ data alone
is not sufficient even with a much larger number of data points compared to $ffCO_2$ data. This is
seen in the estimates of the mean of $ffCO_2$ East Asia and Kor+Jap, where CO pulls this estimate
in the same direction as the ones using $ffCO_2$ data. This adjustment is mostly due to the addition
of model-data error correlation across species ($\mathbf{S_e}$) than source error correlation across species ($\mathbf{S_a}$).
A suggested decrease of CO emissions in East Asia and Kor+Jap, along with an increase in
"Background+non-ffCO" sources resulted to increases in East Asia and Kor+Jap $ffCO_2$ emissions.
Note that our *a priori* HTAPv2 CO and VOC emissions were doubled for East Asia and Korea to
begin with. The slight negative bias in CO at the surface and larger positive bias at 2-3 km,
especially over Seoul and West Sea, is consistent with the adjustments in CO, indicating that bias
in CO is mostly from underestimation of secondary CO and possibly ffCO ROW (e.g., India). The
dominance of $\mathbf{S_e}$ on our results for $ffCO_2$ is in contrast to Boschetti et al. (2018). This may be due
to our approach of localizing our data to below 3 km and aggregating to a smaller number of basis
functions. Nevertheless, *a posteriori* estimates in $ffCO_2$ sources using $ffCO_2$ and CO with $CO_2$
data are statistically significantly indistinguishable from a two-tailed t-test at 99% confidence
interval. This is not the case between *a posteriori* estimates in $ffCO_2$ sources using $ffCO_2$ and $CO_2$
data. We recognize that this is only a proof-of-concept to demonstrate the complementary
information in CO data on $ffCO_2$ at regional scales (even with conservative use of error correlation
estimates). These results are consistent with our analysis of covariation between CO, $CO_2$, and
$ffCO_2$ during the campaign, where the regional difference between air masses from China and
Korea is clearly evident. Vertical profiles of these covariations (both correlation and enhancement
ratio) reveal this regional contrast.
However, the modeled local covariations are confounded by misrepresentation of local and
transport-related processes. Such type of errors can skew the results and have to be addressed (e.g.,
Wu et al., 2018). Our analysis approach was designed to account for these confounding factors
(albeit sub-optimally) by specifying relatively conservative (larger) error covariances and only
using data below 3 km to mimic the sampling distribution of derived $ffCO_2$ measurements, which
is used in this study as our basis of comparison. We are aware that this is still sub-optimal but
detailed refinements to this approach is beyond the scope of this study. We highlight some of these
limitations in Figure 10, where we show vertical profiles of $ffCO_2$ contributions from East Asia,
Kor+Jap and ROW emissions, including the overall bias in $CO_2$ relative to DC-8 $CO_2$ data. While
there is an apparent increase in boundary layer $ffCO_2$ over the West Sea (~1.25 ppm) from the
same increase in *a posteriori* scaling factor relative to *a priori* emissions from East Asia, this
increase only translates to a decrease of ~0.9 ppm in the $CO_2$ bias for this flight group as a result
of all $ffCO_2$ adjustments since there is competing effect between a slight increase in $ffCO_2$ Kor+Jap
and a decrease in $ffCO_2$ ROW. In addition, the use of a single scaling factor for a broad basis
function results to a degradation of $CO_2$ aloft, suggesting that non-$ffCO_2$ and background $CO_2$
needs to be adjusted accordingly by region (not globally) since they are dominant aloft. This
sensitivity between $ffCO_2$ and non-$ffCO_2$ estimates has been pointed out in previous studies (e.g.,
Palmer et al., 2006; Basu et al., 2016; 2020). An added complication to these inversions is the
accounting of $CO_2$ chemical production (Wang et al. 2020) that may also be reflected in the
"Background+non-$ffCO_2$". The aggregation error (Kaminski et al., 2001) confounding our results
also needs to be addressed, perhaps by adding regional basis functions for non-$ffCO_2$ and



background $CO_2$ within a multi-scale (or multi-tiered) hierarchical inversion framework (e.g.,
Cusworth et al., 2020). An ensemble approach using a larger ensemble size from different flux
inversions (e.g., Global Carbon Project, OCO-2 MIP) may offer opportunities to better quantify
the *a priori* error covariances of non-ff$CO_2$ and background $CO_2$. We also recognize that by design
this is a simplistic study focused on CO data as potential constraints on regional ff$CO_2$. A more
realistic scenario would be to show its impact on top of current observational constraints for $CO_2$
(e.g., X$CO_2$ satellite retrievals and derived ff$CO_2$ measurements). Augmenting the flux vector in
$CO_2$ flux inversions with CO and ff$CO_2$ sources may also offer opportunities to understand its
impact on biospheric flux estimates (Basu et al., 2016, 2020; Wang et al., 2020).
There have been several studies using information on local enhancement ($\Delta CO$) that can be derived
from $\Delta CO/\Delta CO_2$ to constrain ff$CO_2$ emissions (Super, 2018). This approach employs assumptions
on the spatiotemporal distribution of emission ratios between CO and ff$CO_2$ using mass balance.
We emphasize here that CO may not be the most appropriate data unless the stationarity
assumption for these $\Delta CO/\Delta CO_2$ are valid and temporal changes in $CO_2$ are reasonably
characterized (e.g., Nassar et al. 2013; Liu et al., 2017). This has been indicated for example in
Super (2018) and Nathan et al. (2018). As has been highlighted in this study, the use of regression
approach in deriving these relationships are confounded by mixing and transport-related processes
making it difficult to attribute the changes in the slopes to emission ratios alone, especially when
analyzing downwind measurements. For this purpose, we suggest a 'model calibration' approach
where ff$CO_2$ emissions are adjusted based on $CO_2$ and CO tags and derived $\Delta CO/\Delta CO_2$ at a
spatiotemporal scale that is representative of the best possible change in combustion efficiency. In
particular, changes in ff$CO_2$ emissions due to changes in CE (through improved technology,
pollution abatement, changes in fuel mixture, process changes, or even decommissioning of a
power plant) do not manifest at local spatiotemporal scale. Ratios derived at finer scale can be
noisy and non-stationary. Changes in emissions due to changes in CE is usually detectable at a far
longer spatiotemporal scale. Long-term satellite retrievals of CO and other proxies of fossil fuel
combustion signatures (e.g., $NO_X$) at decadal timescale (Tang et al., 2019b; Zheng et al. 2018)
may be useful to detect trends on the changes of ff$CO_2$ emissions (Yin et al., 2019).
**7.    Conclusions**
In this study, we highlight the spatial variability of tropospheric CO and $CO_2$ relationships and its
implication in constraining $CO_2$ from fossil fuel combustion. We use the KORUS-AQ field
campaign as our case study. This campaign, which was aimed to study air quality in South Korea,
was conducted on May to June 2016. Incidentally, it also coincided with the peak in global $CO_2$
concentration for this particular year. We use a single-model (CAM-chem) analysis framework,
where the *a priori $CO_2$* fluxes in the model are taken from *a posteriori* fluxes of recent global flux
inversions (e.g., Carbon Tracker – CT2017). We also use CO emissions that were calibrated with
CO data (albeit in an ad-hoc manner) from our previous CAM-chem CO analysis. The availability
of $^{14}CO_2$, CO, and $CO_2$ vertical profiles from NASA DC-8 offers an opportunity to assess the
fidelity of this framework in simulating CO and $CO_2$ abundances from the best possible and
observationally constrained fluxes and emissions. More importantly, this framework enables us to
facilitate a better understanding of the variability in observed and modeled relationships between
the abundances of these species. Our analysis is directed towards investigating the covariation of
CO, $CO_2$, and ff$CO_2$, which can then be made useful in refining our estimates of regional ff$CO_2$
emissions.


We evaluated CAM-chem CO and $CO_2$ simulations from a variety of observing system perspectives, while focusing on key diagnostics relative to KORUS-AQ measurements and previous model and data analysis for this particular period and region. Our results show that the spatiotemporal distribution of CAM-chem CO and $CO_2$ simulated abundances (and their associated correlations and enhancement ratios) are reasonably consistent (and within the range of uncertainties) with KORUS-AQ CO and $CO_2$ data, CAMS high resolution forecast/analysis of CO and $CO_2$, and CT2017 mole fractions for $CO_2$ -- both of which used different transport models at different resolution. In particular, we find that: 1) The overall biases against DC-8 $CO_2$ and CO measurements in CAM-chem using CT2017 fluxes are -1.0 ppm and -24 ppb, respectively, while the CAMS FC9s is biased by about 0.7 ppm in $CO_2$ and -17 ppb in CO. The CT2017 $CO_2$ mole fraction is biased by -1.2 ppm; 2) The overall correlation ($R_{CO,CO_2}$) and enhancement ratio ($\Delta CO/\Delta CO_2$) between CO and $CO_2$ are as follows: DC-8: 0.67 and 13.3±0.21 ppb/ppm, CAM-chem: 0.55 and 13.8±0.23 ppb/ppm, and CAMS FC9s – 0.65 and 12.5 ppb/ppm. The error correlation $errR_{CO,CO_2}$ in CAM-chem (0.40) is also comparable to CAMS FC9s (0.49); 3) The overall bias in CAM-chem ff$CO_2$ against $^{14}CO_2$ data is -1 ppm, which is close to 1-*sigma* uncertainty of the data (1 ppm). We also note that the modeled CO and $CO_2$ correlation and enhancement ratios vary differently relative to DC-8, suggesting possible misrepresentation of related sources and sinks in CAM-chem. In particular, we find a significantly lower (higher) correlation near the surface (aloft) over West Sea relative to DC-8, whereas its enhancement ratio is comparable near the surface but larger aloft. We attribute this difference to coarser representation of boundary layer processes (low correlation) and overestimation of regional emission ratio aloft (high enhancement ratio).

We also investigated the contribution of regional ff$CO_2$ to observed $\Delta CO/\Delta CO_2$ using tagged ff$CO_2$ simulations. We find that, even near the surface in Seoul, there is a significant contribution of background and non-ff$CO_2$ that cannot be neglected. Its median contribution across flight groups is 74% below 1.5 km, 47% between 1.5 and 3km and 81% > 3 km. ff$CO_2$ from East Asia also contributes significantly, with median contributions ranging from 10% below 1.5km, 35% between 1.5 and 3 km, and 20% >3 km. Its higher contribution is especially evident at all levels over the West Sea air samples, which are representative of Chinese pollution outflows. These variations in contributions affect the design and interpretation of joint CO:$CO_2$ inversions. We find, for example, that in order to effectively constrain ff$CO_2$ emissions from Kor+Jap and East Asia, we have to localize our inversion to data points below 3 km. Else, the larger impact of "Background+non-ff$CO_2$" can obscure the response from ff$CO_2$ emissions. We conducted three sets of inversions to demonstrate the impact of CO data in refining estimates of regional ff$CO_2$ emissions. While recognizing the simplicity of our joint Bayesian synthesis inversion (which follows Palmer et al., 2006), we find that ff$CO_2$ from East Asia and Kor+Jap need to be increased by 27%±24% and 9%±17%, respectively. This is very consistent (albeit with larger uncertainty) with results from an inversion using derived ff$CO_2$ data only (East Asia: 27%±9% and Kor+Jap: 10%±3%). In contrast, inversion using only $CO_2$ data results to a decrease in both East Asia (-5%±27%) and Kor+Jap (-6%±19%) reflecting the difficulty to differentiate the response of background+non-ff$CO_2$ and regional ff$CO_2$ using $CO_2$ profiles alone.

Although these results are promising, we emphasize that this is only proof-of-concept which needs to be refined with more rigorous and realistic inverse modeling experiments for different observing systems. This is especially the case for global inversion systems that take into account the appropriate scales inherent in these types of information and goes beyond the use of traditional



error covariance estimation. CO, in particular, is useful in constraining ffCO$_2$ at regional scales
since this scale is commensurate to its lifetime of 1 to 2 months. It becomes problematic at local
scales due to its inherent confounding factors and inability of global chemical transport models to
capture its variability at these scales. While this study focuses on a specific region, we highlight
in this work the importance of rigorously verifying the relationships and sensitivities derived from
regional and global models to any joint inverse analyses. It is especially important to verify
consistencies across species. Careful consideration of associated errors on the vertical distribution
of these sensitivities and assumptions of stationarity is warranted, especially for future joint
analyses using satellite columnar retrievals of these species, which lack vertical information and
may not necessarily be collocated in both space and time.



**Acknowledgments**

This study is supported by NNX16AE16G, NNX17AG39G and NNH18ZDA001N. We also thank the CESM and CAM-chem team for technical support, including Stephanie Wuerth for sharing her CAM/DART code modifications. The CESM project is supported primarily by the National Science Foundation (NSF). This material is based upon work supported by the National Center for Atmospheric Research, which is a major facility sponsored by the NSF under Cooperative Agreement No. 1852977. Computing and data storage resources, including the Cheyenne supercomputer (doi:10.5065/D6RX99HX), were provided by the Computational and Information Systems Laboratory (CISL) at NCAR. We especially acknowledge the scientific and product teams in CarbonTracker, CarbonTracker-Europe, and CAMS GHG inversion for the $CO_2$ flux products that they have kindly provided. CarbonTracker CT2017 results provided by NOAA ESRL, Boulder, Colorado, USA from the website at http://carbontracker.noaa.gov. We also thank the teams involved in HTAP and FINN for CO emission inventories. We acknowledge the following teams for their great effort in taking $CO_2$ and CO measurements and providing them publicly: KORUS-AQ for DC-8 measurements, TCCON for XCO and $XCO_2$, NOAA ESRL Carbon Cycle Cooperative Global Air Sampling Network for the surface air flask sampling data, MOPITT and OCO-2 for XCO and $XCO_2$ retrievals, respectively. We thank Dr. Donald Blake's research group from the University of California, Irvine for collecting the airborne flask samples. The NCAR MOPITT project is supported by the National Aeronautics and Space Administration (NASA) Earth Observing System (EOS) Program. We specially thank Dr. Frédéric Chevallier and Dr. Britton Stephens for their insightful comments on improving this manuscript. All the fluxes and emissions, and observational data are available online.

**Code and datasets**

CESM2.0 is a publicly released version of the Community Earth System Model and freely available online (at www.cesm.ucar.edu, last access: 14 August 2020). The Korea-United States Air Quality Field Study (KORUS-AQ) dataset is available at https://doi.org/10.5067/Suborbital/KORUSAQ/DATA01. MOPITT data is available at https://www2.acom.ucar.edu/mopitt while the Orbiting Carbon Observatory-2 $XCO_2$ is available at https://disc.gsfc.nasa.gov/datasets/OCO2_L2_Lite_FP_9r/summary. The Total Carbon Column Observing Network (TCCON) and NOAA datasets can be downloaded at https://tccondata.org and (https://www.esrl.noaa.gov/gmd/ccgg/flask.php), respectively.



**Appendix A.  Tagging ffCO$_2$ and ffCO in CAM-chem**
The abundance of tropospheric $CO_2$ at any given space ($s$) and time ($t$) can be decomposed into
contributions from different processes. That is,

$CO_2(s,t) = CO_2^{bg}(s,t)$
$\qquad + \left(CO_2^{ffbf}(s,t) + CO_2^{bb}(s,t) + CO_2^{cem}(s,t) + CO_2^{res}(s,t) + CO_2^{chem}(s,t)\right)$
$\qquad - \left(CO_2^{lnd}(s,t) + CO_2^{ocn}(s,t) + CO_2^{st}(s,t)\right)$ $\qquad\qquad$ (A.1)

where $bg$ denotes background, $ffbf$, $bb$, $cem$, $res$ and $chem$ are $CO_2$ sources from fossil
fuel/biofuel combustion, biomass burning, cement production, biospheric respiration, and
chemical production processes, while $lnd$, $ocn$, $st$ are $CO_2$ sinks due to biospheric
(photosynthetic) uptake, ocean-tropospheric, and tropospheric-stratospheric exchange,
respectively. Our notation of non-ffCO$_2$ corresponds to other sources that are not $ffbf$.

Similarly,

$CO(s,t) = CO^{bg}(s,t)$
$\qquad + \left(CO^{ffbf}(s,t) + CO^{bb}(s,t) + CO^{oxid}(s,t)\right)$
$\qquad - \left(CO^{OH}(s,t) + CO^{dep}(s,t)\right)$ $\qquad\qquad$ (A.2)

where $oxid$, $OH$ and $dep$ denote secondary CO due to VOC oxidation, CO sinks due to its
reaction with $OH$ radical and dry deposition, respectively.

We have developed tagging capabilities in CAM-chem for both CO and CO$_2$ sources by
prescribing their associated sinks. Tagging CO has been developed in the past by treating CO from
a particular basis function as tracers. That is, we solve the continuity equation for every tagged CO
in the same way as the default CO variable in the model but making sure that each tagged CO does
not interact with model chemistry (i.e., by treating it as a passive tracer). This mechanism is
mentioned in Emmons et al. (2012) and previously used in Bayesian synthesis inversion studies
(e.g., Arellano and Hess, 2006) and chemical budget studies (Gaubert et al., 2016). A similar
approach is also used by Fisher et al. (2017) with GEOS-Chemv9 model. This tagging capability
is further illustrated in Eq. A.3 for a particular tag CO ($itag$).

$\dfrac{\partial [X]^{itag}}{\partial t} = \left.\dfrac{\partial [X]^{itag}}{\partial t}\right|_{transport} + \left.\dfrac{\partial [X]^{itag}}{\partial t}\right|_{sources} - \left.\dfrac{\partial [X]^{itag}}{\partial t}\right|_{sinks}$ $\qquad\qquad$ (A.3)

The temporal evolution of a tracer $[X]^{itag}$ for each grid cell in the model is calculated using the
same continuity equation for species $[X]$. As expressed in Eq A.2, this includes the background
dynamics represented here as transport term (dynamics and physics incl. advection, diffusion,
mixing, convection, and CO flux convergence and divergence), all sources (emissions and
chemical production), and all sinks (CO+OH reaction, and deposition). These tags or basis can be
either disaggregated sectoral components and/or regional source components of CO depending on
the problem to be addressed. Here, we use ffCO emitted from a few regions around Korea as our





basis. All these regions are defined in Figure 1. The response of this basis or the contribution of this source region to overall abundance in CO is estimated by integrating Eq. A.3. Hence, the simulated $[CO]^{itag}$ for example corresponds to $[CO]$ mixing ratio for a given mass of CO emitted to the atmosphere by this *itag* region. The CO tags added in CAM-chem consists of the following edits to the code: (1) The CO tags are defined in the chemical preprocessor (variable names are arbitrary defined as "CO01", "CO02" …); (2) emission files for the tags of emissions from specific regions are prepared and defined in the namelist; (3) chemical production of CO for CO tags of chemical sources are defined by adding related chemical reactions in chemical preprocessor; (4) the OH chemical loss is defined in the chemical preprocessor, OH is not affected by the oxidation of tags; (5) dry deposition for the CO tags is applied in the same way as for the default CO variable. Detailed evaluation and validation of CAM-chem CO tags can be found in Tang et al. (2019a) and https://wiki.ucar.edu/display/camchem/.

We apply a similar approach in tagging ffCO$_2$ (Eq. A.1 and Eq. A.3). However, we do not account for chemical production in the source term nor deposition in the sink term. The sink of each ffCO$_2$ tags is derived from the negative surface flux $f_{CO_2}^{itag}$, which we define as the product of the negative surface flux of CO$_2$ ($f_{CO_2}$) at a given time and the ratio of the associated CO$_2$ mixing ratio of the tag ($[CO_2]_{srf}^{itag}$) at the surface and the modeled CO$_2$ mixing ratio $[CO_2]_{srf}$ at the surface; i.e.,

$$f_{CO_2}^{itag} = f_{CO_2} \cdot \left( [CO_2]_{srf}^{itag} \Big/ [CO_2]_{srf} \right) \tag{A.4}$$

In this manner, the sink of model CO$_2$ can be disaggregated into the sum of the sinks for all tags. This ensures that the relative abundance of the tagged CO$_2$ to the total CO$_2$ is conserved. Other sources of CO$_2$ (chemical oxidation) is treated as part of the "Background+non-ffCO$_2$" in the same manner as the secondary CO within "Background+non-ffCO". Edits to the model include: 1) The CO$_2$ tags are defined in the chemical preprocessor similarly as "CO2_online" (named "CO2_online_anthro", "CO2_online_fire", "CO2_online01", "CO2_online02", …); (2) positive flux (source) files for the tags from specific regions are prepared and defined in the namelist; (4) sinks of all tags are defined using Eq. A.4. The routines, mo_srf_emissions.F90 and chemistry.F90 codes of the CESM chemistry routines are modified for this development. The modified CAM-chem source codes and chemical preprocessor are accessible through Github (See data availability for details).



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



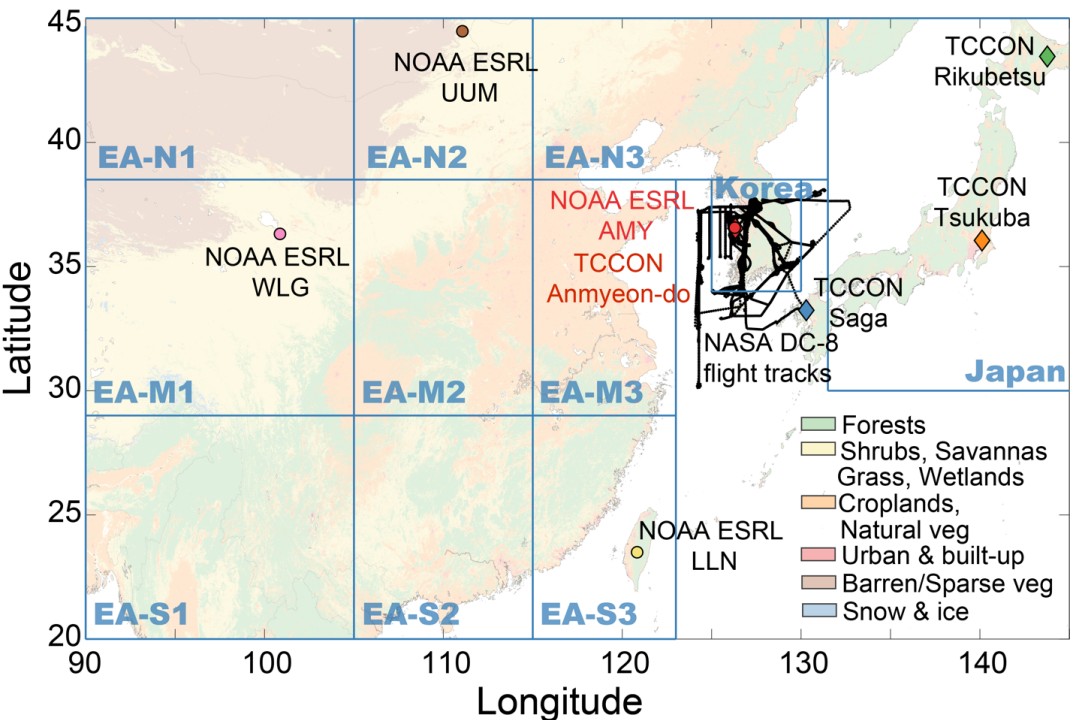

**Figure 1.** Map of the study domain including: land cover (colored map), definition of tag (basis) regions (blue rectangles), location of four East Asia sites from the NOAA ESRL Carbon Cycle Cooperative Global Air Sampling Network (colored dots), location of East Asia TCCON sites (colored rhombus), and the DC-8 aircraft flight tracks during KORUS-AQ (black lines).

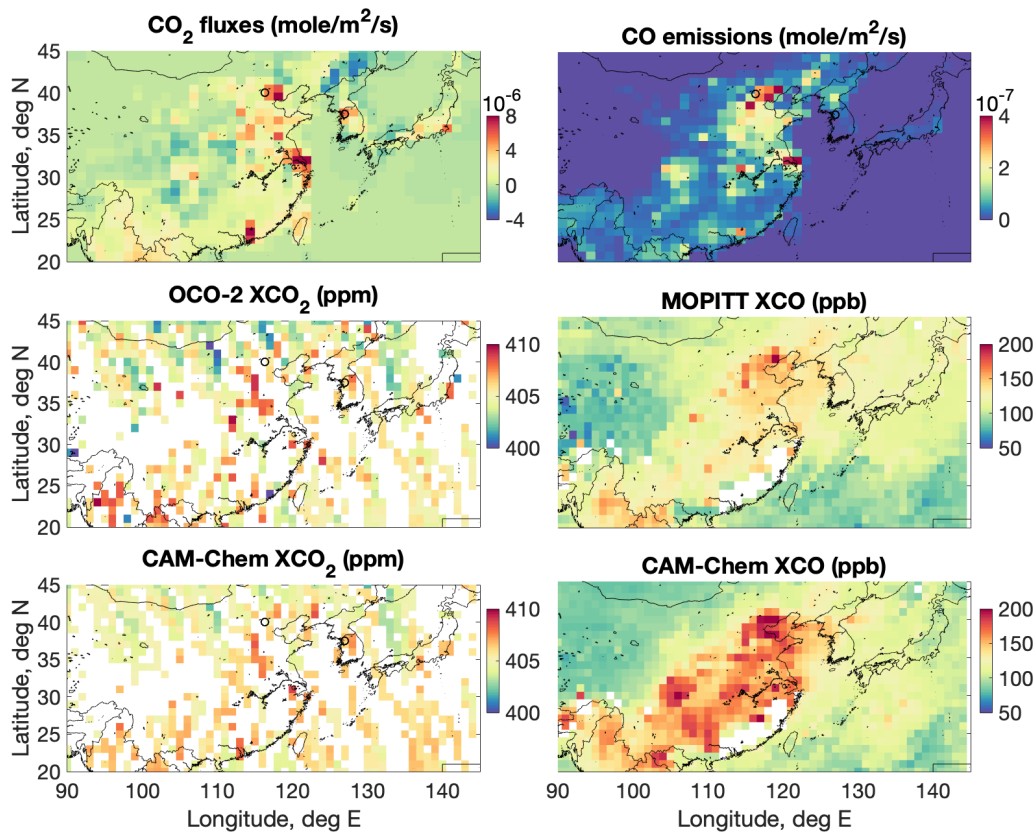

**Figure 2.** Spatial distribution of *a priori* mean $CO_2$ fluxes from CT3h (top left), CAM-Chem CO emissions (top, right), OCO-2 $XCO_2$ (middle left) and MOPITT XCO composites (middle right) for the entire KORUS-AQ campaign period. Also shown is the spatial distribution of CAM-Chem $XCO_2$ (bottom left) and XCO (bottom right) model equivalents. See Figure S5 for sub-monthly comparisons.



**Figure 3.** Campaign composite of KORUS-AQ DC-8 flight $CO_2$ (a) and CO (b) data, model equivalent $CO_2$ from CAM-Chem (c) and CO (d), and $CO_2$ from Carbon Tracker (CT2017) $CO_2$. Panel f) shows the flight tracks for the flight groupings in this study.

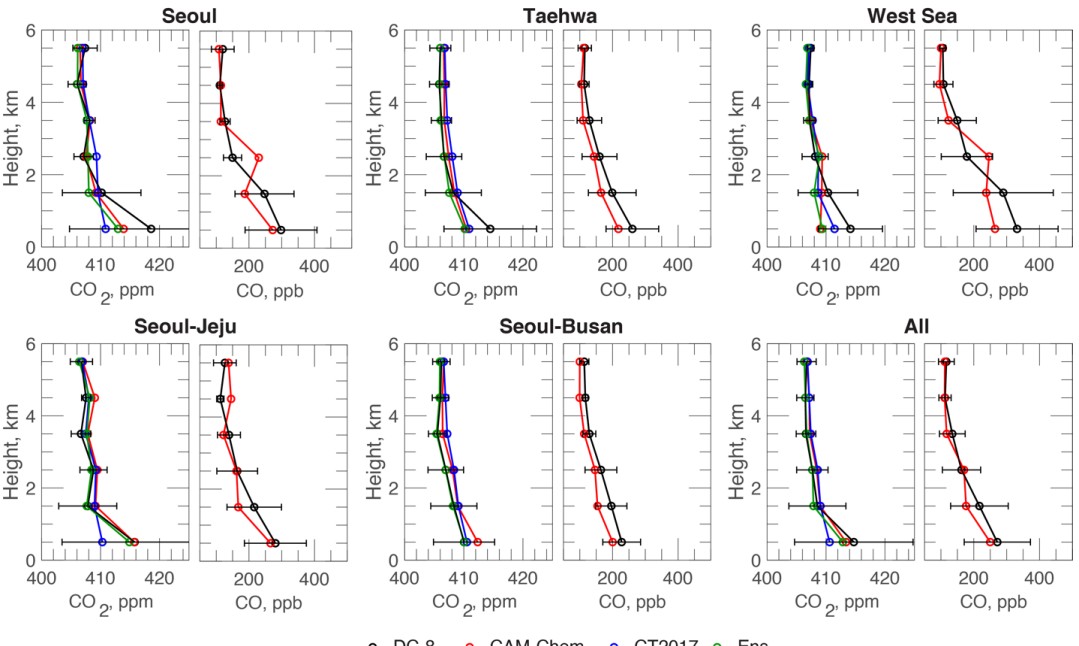

**Figure 4.** Mean vertical profiles of $CO_2$ (ppm) and CO (ppb) averaged across the KORUS-AQ campaign period by flight groups (see Figure 3f for the location of these groups). DC-8 data $CO_2$ and CO are shown in black (with error bars corresponding to its standard deviation). Superimposed are model equivalents of $CO_2$ and CO from CAM-Chem (red), $CO_2$ from Carbon Tracker (CT2017, blue), and ensemble mean $CO_2$ from CAM-Chem using CT3h, CAMS, and CTE2018 fluxes (green).



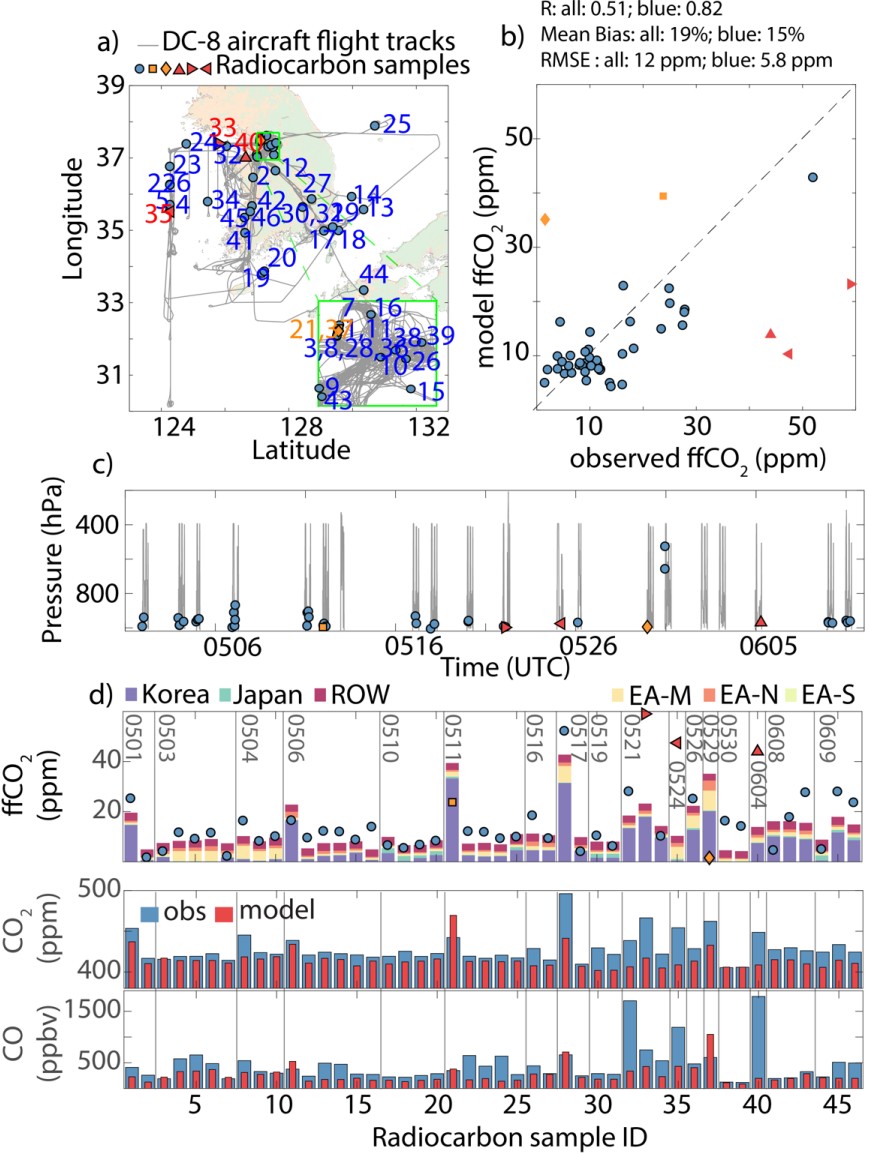

**Figure 5.** Comparison of ffCO$_2$ with radiocarbon ($^{14}$CO$_2$) data during KORUS-AQ. The spatial and temporal sampling of $^{14}$CO$_2$ (colored markers) and CO$_2$ measurements (gray line) are shown in top left panel (a), (horizontal) and middle panel (c) (vertical and time), respectively. Data points colored in orange and red are considered outliers. The top right panel (b) correspond to a scatterplot between ffCO$_2$ from CAM-chem tags and ffCO$_2$ from $^{14}$CO$_2$ (overall correlation is indicated for all data points and excluding outliers). Modeled regional contributions to ffCO$_2$ are shown in the bottom panel (d) along with the values of $^{14}$CO$_2$ samples (ppm), and observed and modeled CO and CO$_2$ in the bottom panels of d).





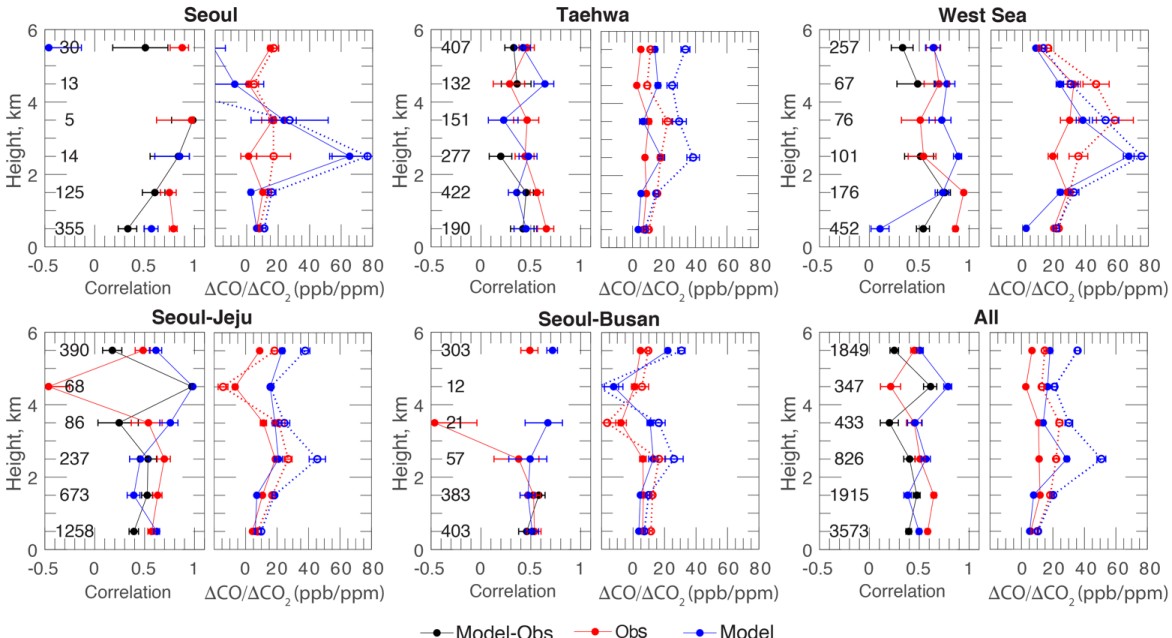

**Figure 6.** Vertical profiles of mean $CO:CO_2$ correlations (left panels) from DC-8 (red) and CAM-Chem/CT3h (blue), and the correlation between model CO minus DC-8 CO and model $CO_2$ minus DC-8 $CO_2$ (black) arranged by flight groups. Right panels correspond to vertical profiles of derived enhancement ratios ($\Delta CO : \Delta CO_2$) from DC-8 (red) and CAM-Chem/CT3h (blue) based on ordinary least squares (OLS) regression. Open circles with dotted lines are enhancement ratios derived using reduced major axis (RMA) regression at $p<0.05$. Number of data points for each vertical layer (1-km) bin is shown in the left panels. The error bar denotes the associated uncertainty of every estimate. Missing values denote non-statistically significant ($p<0.05$) correlations.



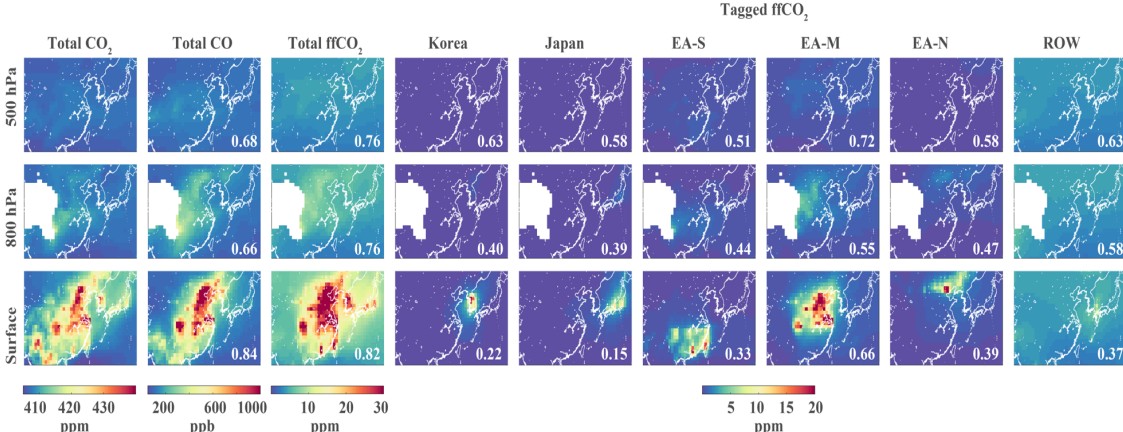

**Figure 7.** Spatial distribution (averaged across KORUS-AQ) of modeled total $CO_2$ (ppm) and CO (ppb), modeled $ffCO_2$ and $ffCO_2$ tags at model surface, 800 hPa, and 500 hPa. Pearson (pair-wise) correlation coefficients across the domain relative to total $CO_2$ are shown in the bottom right of each image.





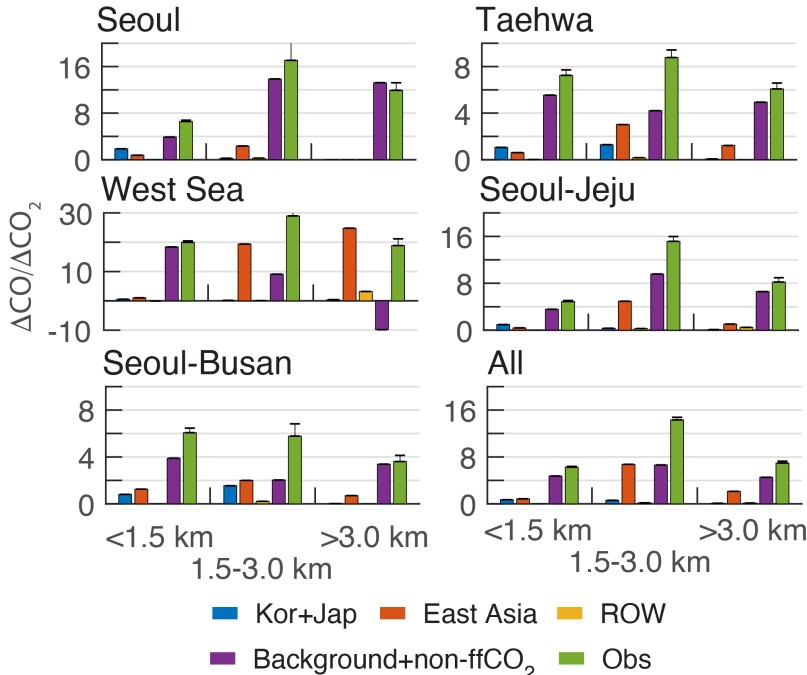

**Figure 8.** DC-8 ΔCO: ΔCO₂ (green) and associated uncertainty (error bar) derived from all data points within a flight group and vertical layer (0 to 1.5km, 1.5-3.0km and >3.0km). Also shown are contributions of each optimized response functions (based on an inversion using ffCO₂ data, see Figure 9) to the overall observed sensitivity.





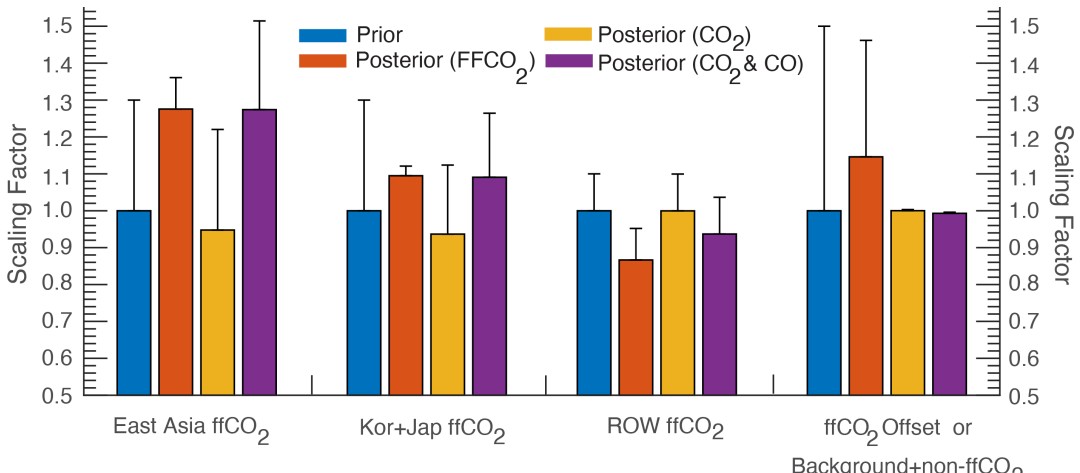

**Figure 9.** *A priori* (blue) and *a posteriori* estimates of ffCO$_2$ scaling factors (and associated uncertainty shown as an error bar) from a Bayesian synthesis inversion using ffCO$_2$ data derived from $^{14}$CO$_2$ samples (red) and inversion using DC-8 CO$_2$ (yellow-orange) and joint inversion using DC-8 CO$_2$ and CO (magenta). Here, the basis functions are aggregated to include East Asia, Kor+Jap, Rest of the World ffCO$_2$ and "ffCO$_2$ offset" (for ffCO$_2$ inversion) or "Background+non-ffCO$_2$" (for CO$_2$ or CO$_2$ and CO inversions).



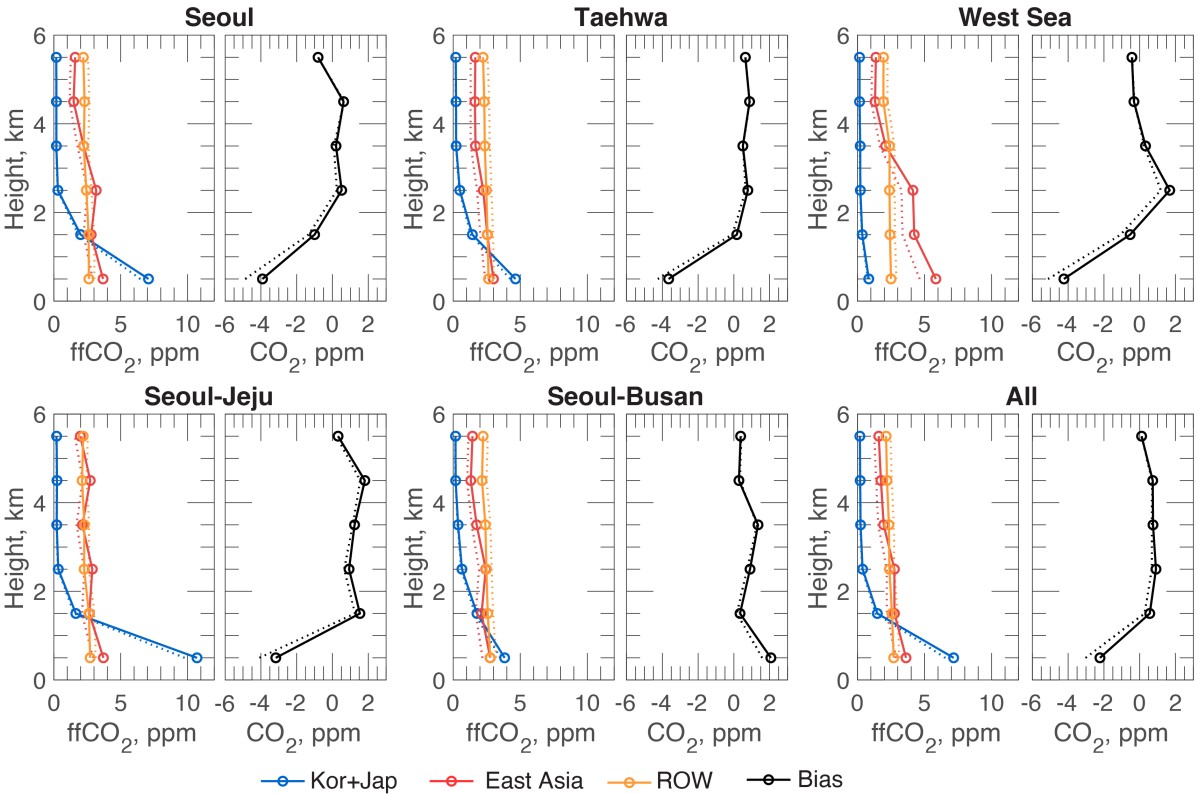

**Figure 10.** Mean vertical profiles of ffCO$_2$ response functions from Kor+Jap (blue), East Asia (red) and ROW (yellow-orange) for each flight group. Dashed and solid lines correspond to *a priori* and *a posteriori* estimates, respectively. Mean CO$_2$ bias (model-obs) are shown in the right panels.





**Table 1**. $CO_2$ fluxes used in this study.

| $CO_2$ fluxes | Spatial Res. | Temporal Res. | Period | Transport Model | Fossil Fuel Priors | Biosphere and Fires Priors | Ocean Priors | Main Reference |
|---|---|---|---|---|---|---|---|---|
| CT2017 | 1º lon 1º lat | 3-hourly | 2000-2017 | TM5 | "Miller" (EDGAR scaled to CDIAC) & "ODIAC" | CASA w/ GFED 4.1s GFED_CMS | Jacobson et al. (2007) Takahashi et al (2009) | Peters et al. (2007)[1] |

[1]With updates documented at http://carbontracker.noaa.gov.





**Table 2.** Summary statistics of CO and $CO_2$ NASA DC-8 measurements. npair is the number of data pairs of CO and $CO_2$. Model equivalents and model evaluation against CO and $CO_2$ data are also shown. Units are ppm for $CO_2$ and ppb for CO.

| | | All | Seoul | Taehwa | West Sea | Seoul Jeju | Seoul Busan |
|---|---|---|---|---|---|---|---|
| npair | | 8942 | 542 | 1579 | 1129 | 2712 | 1179 |
| Obs Mean | $CO_2$ | 410 | 415 | 408 | 411 | 411 | 408 |
| | CO | 205 | 266 | 163 | 234 | 223 | 183 |
| Obs Std | $CO_2$ | 7.7 | 13 | 5 | 5 | 10 | 4 |
| | CO | 101.9 | 113 | 73 | 143 | 101 | 64 |
| Obs $R_{CO2,CO}$ | | 0.66 | 0.79 | 0.68 | 0.89 | 0.62 | 0.60 |
| Obs $\Delta CO/\Delta CO_2$ | | 13.30 | 9.13 | 15.28 | 28.20 | 10.37 | 15.92 |
| Model Mean | $CO_2$ | 410 | 412 | 408 | 409 | 412 | 410 |
| | CO | 188.4 | 237 | 143 | 202 | 213 | 155 |
| Model Std | $CO_2$ | 7.8 | 10.5 | 4.2 | 3.5 | 10.1 | 5.8 |
| | CO | 107.1 | 133 | 70 | 119 | 117 | 62 |
| Model $R_{CO2,CO}$ | | 0.55 | 0.59 | 0.50 | 0.39 | 0.67 | 0.60 |
| Model $\Delta CO/\Delta CO_2$ | | 13.80 | 12.61 | 16.56 | 33.66 | 11.54 | 10.68 |
| Bias Model minus Obs | CT3h | -1.0 | -3.5 | -0.1 | -2.2 | -1.4 | 0.8 |
| | CT2017 | -1.2 | -3.5 | -0.4 | -1.3 | -1.9 | 0.6 |
| | CO | -24.2 | -29.2 | -20.4 | -32.6 | -34.5 | -27.9 |
| R Model versus Obs | CT3h | 0.39 | 0.60 | 0.45 | 0.40 | 0.38 | 0.05 |
| | CT2017 | 0.37 | 0.43 | 0.60 | 0.51 | 0.32 | 0.21 |
| | CO | 0.63 | 0.63 | 0.64 | 0.67 | 0.59 | 0.72 |
| RMSE | CT3h | 7.7 | 11.0 | 4.7 | 5.3 | 9.3 | 7.0 |
| | CT2017 | 6.9 | 9.8 | 3.9 | 4.6 | 9.0 | 4.8 |
| | CO | 87.6 | 111.5 | 64.0 | 113.6 | 90.3 | 55.2 |
| errorR | CT3h | 0.40 | 0.36 | 0.41 | 0.57 | 0.41 | 0.43 |