# Peer review of "On the relationship between tropospheric CO and CO2 during KORUS-AQ and its role in constraining anthropogenic CO2"

_Atmospheric Chemistry and Physics, 2020_

## Referee Comment (RC1) · Anonymous Referee #1 · 4 Nov 2020

The manuscript "On the relationship between tropospheric CO and CO2 during KORUS-AQ and its role in constraining anthropogenic CO2" by Tang et al. presents analysis based on the CO, CO2 and 14CO2 data collected during the KORUS-AQ campaign over South Korea. It compares simulations of CO, CO2 and FFCO2 concentrations from a global transport model to the data and presents inversions of the FFCO2 emissions in East Asia (Eastern China, Korea and Japan) using this transport model and these data.

The prospect of the joint analysis of CO2, CO and 14CO2 data supported by transport model simulations and of their joint assimilation in an inversion system is very

promising. Sometimes, the manuscript nearly reaches interesting insights on this topic. However, in a general way, the study and the manuscript fail to exploit the potential of such analysis. I think that more work and thoughts are needed to produce a paper that deserves publication and that this goes beyond what is usually done for "major revisions".

Here are some of my main concerns regarding this manuscript:

1) It is often very difficult to follow because the writing and the reasoning are not structured and rigorous enough. A large amount of sentences are confusing because of the lack of clarity, precision and explanations. The reading of the long series of statistics lacks of hierarchy. The use of simulations or datasets that are not much exploited in the analysis does not help. For instance, efforts are needed to follow, in section 3, whether or why under or over estimations of concentrations are supposed to highlight the under or over estimation of local, regional or global sources and sinks. The reading of section 4 is even more difficult.

Furthermore, despite being relatively long, the text goes too fast on some of the critical parts of the reasoning like the rationale for the study in the introduction and the justification for the use of specific inversion configurations and parameters.

I think that many assumptions and analysis are debatable and that the manuscript shows a lack of hindsight on the topic and results of this study.

2) The specific scope and the objectives of sections 3 and 4 are not clear. This manuscript is the 5th one analyzing the $CO_2$ and/or CO data from the KORUS-AQ campaign using transport models (after Tang et al. 2018, Halliday et al. 2019, Tang et al. 2019a and Gaubert et al. 2020, ACPD). How do section 3 and 4 draw on these previous publications and bring new learning ?

Furthermore, opposed to what is claimed repeatedly (e.g. in the title, the abstract, the beginning of section 1.1, the beginning of the conclusion...), I hardly see how

these sections feed the configuration (in particular the error covariance matrices) or the analysis of the inversions in section 5.

The introduction does not help much. Until the beginning of section 1.1, this is a collection of very general (and sometimes misleading) statements about the monitoring CO2 anthropogenic emissions using atmospheric data. Section 1.1 fails to bring clear specific context, rationale and objectives for this study. Lines 150-154 look like a summary of the activities rather than a list of objectives. Lines 154-172 attempt to distinguish this new study from the previous ones by pointing to practical differences that are hardly convincing. Lines 769-770 and 775-776 claim that the set-up of matrix Se follows values of errors on CO and CO2 and errRCO,CO2 from section 4 but I do not see how. Anyhow, the errors and the correlation of errors in CO and CO2 modeled concentrations are driven by the atmospheric transport, the surface fluxes (and other source and sinks), and to errors in both the transport model and in the modeled sources and sinks. I do not see why it should be used to characterize transport model errors only. The authors seem to miss the links between their statistics of model-data differences and Sa (see below my general comment about the key role that this matrix should have played).

3) There is a critical lack of proper discussion on the spatial extent and scales that are suitable for the analysis of the data. Parts of sections 3 to 5 attempt to distinguish the influence of Eastern Asia or the rest of the world vs that of Korea, l340-350 discuss the sources overflown during the campaign and the flights over the West Sea are said to be designed to capture "China pollution outflow". Some sentences even raise (too late) some concerns associated to the coarse spatial resolution of CAM-chem. However, in general, and in particular in the title, introduction and section 2, there is no real reasoning regarding the observation footprints and regarding the modeling and inversion domain and resolution that are suited for these data.

The introduction mixes all inversion scales and all types of observation networks. Nothing is said about the wind fields during the campaign. The comparisons of a single

global coarse resolution model to TCCON individual sites, to OCO-2, to MOPITT data or to the aircraft data are brought together without consideration for the differences between these observation datasets in terms of flux and process representativeness. The statistics obtained here are sometimes compared with results from other campaigns in different regions or with different coverage, or from "state-of-the-art" models (without mentioning whether they are global or regional). The contrast between the spatial extent of the regions for which the total emissions are rescaled by the inversion and the local (nearly vertical) reading of the data footprints at l340-350 is questioning.

Given the spatial extent of the KORUS-AQ campaign (less than 8°x8°) and its density of data over South Korea, the use of a global 1° resolution transport model with coarse vertical resolution to analyze it is not obvious and sounds like a step backward compared to the previous publications on this campaign (which used higher resolution models). From what I understand, the model is interpolated at each observation location and all the statistics are derived over the ensemble of observation locations. What can be the meaning of such statistics at a resolution much finer than that of the model ?

4) The usual concept for the co-assimilation of CO and CO2 is that the signal from misfits between modeled and measured CO can be used to add constraint on the inversion of FFCO2 emissions because uncertainties in the FFCO2 emissions are connected to uncertainties in FFCO emissions. This usually translates into two options for the inversion: (a) rescaling activity levels underlying both FFCO and FFCO2 emissions rather than FFCO and FFCO2 emissions separately or (b) rescaling separately FFCO and FFCO2 emissions accounting for positive correlations between their respective prior uncertainties. The key challenge for joint CO-CO2 inversions is usually thought to be the uncertainties and high spatial and temporal variations in the CO/CO2 emission ratios i.e. in (a) the uncertainties and variations in the CO and CO2 emission factors to be multiplied by the common activity indices to get emissions and in (b) the level and variations of the positive correlation between the prior uncertainties in FFCO and

FFCO2 emissions. The authors of this study control separately FFCO and FFCO2 emissions but neglect the correlations between their respective prior uncertainties or acknowledge that they have no idea about how to parameterize them, cutting the critical connection between FFCO2 and FFCO (see lines 819-830). The configuration of Sa is actually made all the more difficult by using two different inventories for prior FFCO2 and FFCO emissions, which undermines the ability to rely on tights connections between these emissions (especially since the CO inventory has been multiplied by 2 to better fit the data before the inversion).

The authors even assume that these correlations are negative, i.e. that the combustion efficiency could be the main source of uncertainty in both FFCO2 and FFCO emissions. However, the prior FFCO and FFCO2 emissions are based on different inventories (and FFCO emissions have been multiplied by 2), so that, in principle, the ratios between these emissions should not correspond to an assumption on this efficiency. In a more general way, such an assumption is quite surprising. Uncertainties in the level of activity at national, and even more at regional to local scales should be a dominant source of uncertainty in FFCO2 emission inventories (especially at sub-annual temporal scales). Such a driver of uncertainty in FFCO2 emissions raises correlations with uncertainties in FFCO emissions. Even if uncertainties in CO emission factors are one of the main sources of uncertainties in FFCO emissions, their counterpart in FFCO2 emissions (the generation of plus or minus CO2 depending on the combustion efficiency) can hardly balance this driver. The assumption of negative correlations between uncertainties in FFCO2 and FFCO prior emissions is further weakened by various discussions in sections 3 and 4 that point to common underestimation or overestimation of sources in both the CO2 and CO inventories.

The idea that Se could ensure the expected connection between FFCO2 and FFCO is not relevant. Se prevents the inversion from overfitting the data and limits the corrections to the prior emissions. Correlations between CO2 and CO transport modeling errors help the inversion better filter such errors when deriving FFCO2 and FFCO emissions but this has a very indirect and weak impact on the connection between these emissions. One could think that the results from the inversions in section 5 contradict these statements. However, I assume that the consistency between FFCO2 and CO2-CO inversions and their divergence from CO2 inversions is intrinsically linked to the very low dimension of these inversions (with 4 or 8 unknowns). Various assumptions and settings of these inversion raise issues (see below) and I thus have the feeling that this result is not robust.

The general topic of the CO-CO2 correlations that should be at the heart of this paper is complex and I feel that the authors missed it.

5) Some concerns about the inversion configuration:

- I don't understand how global inversions (even if they focus on East Asia) assimilating CO2 (with or without CO) can behave correctly when inverting scaling factors for FFCO2 emissions only and keeping other CO2 fluxes fixed. The "posterior" biogenic fluxes from CTE and CAMS bear large uncertainties. The problem is enhanced by the lack of spatial resolution of the inversion, which does not help distinguish the patterns from anthropogenic vs biogenic fluxes in the data.

- I do not understand the rationale behind removing observations above 3 km (the "localization purposes": l791-792 refer to "previous section" but I do not see where sections 3 and 4 help understand this choice). The inversions are global, and data above 3 km should be helpful for constraining ROW.

- I have some doubts regarding the way the FFCO2 simulations are compared to gradients of FFCO2 from gradients in 14CO2 observations: I believe that the authors should have computed, in the FFCO2 simulations, the gradients to the location of the 14CO2 background sites rather than introduce a "ffCO2 offset".

6) There is a lack of consideration for the inventories. As mentioned above, the fact that the FFCO2 and FFCO inventories are quite independent strongly hampers the interpretation of the model-data comparisons and the reading of CO/CO2 ratios in the model. The EDGAR inventory, which plays a critical role in this study (this is the FFCO2 emission inventory behind the simulations in sections 3 and 4 and the prior emission inventory in section 5) is not even named in the main text, the authors speaking about CTE and CAMS fluxes (which combine natural fluxes optimized through global inversions with the EDGAR inventory) only. Differences between the CAMS analysis and CAM-Chem simulations are not supported by insights on the emission maps behind these two simulations. The FFCO2 emissions from the inversion are hardly compared to official inventories (and the CO emissions from the inversions are not discussed).

7) I could draw a long list of secondary issues in specific paragraphs or sentences (and even in the mathematical notations and equations) but I restrain myself to the main ones raised above

In conclusion, I think that the scope and configuration of the analysis should be rethought, and that the presentation of such a study should be strongly improved. I thus recommend this manuscript to be rejected and re-submitted after a deep revision.

---

## Referee Comment (RC2) · Peter Rayner (Referee) · 9 Nov 2020

**General Comments**

this paper examines the role of carbon monoxide (CO) measurements in a joint inversion of CO and CO$_2$ fluxes over East Asia. The paper builds on methodologies and results from some previous papers but delves further into the specific role of CO in improving the inversion.

I found the paper difficult to assess mainly because I have never felt comfortable that I understand the methodology of Palmer et al. (2006). It's quite possible therefore that

what I'm about to say is wrong. I also want to add that I am taking advantage of the discussion period to comment on the paper, i.e. this is not a review.

As I understand it the Palmer et al. (2006) methodology defines independent Jacobians for CO and $CO_2$ i.e. there is no physical coupling between the two sources or between each source type and the other observation type. These relationships are introduced by correlations in the observation and prior covariance matrices. It's obvious this works for prior correlations and less obvious (though still true) that it works for observations.

It is worth comparing the approach with what happens in nature. Sources are clearly related since some $CO_2$ fluxes are the result of combustion which may also produce CO with an uncertain yield. Writing the problem as $F_{CO} = \alpha F$
$F_{CO2} = (1-\alpha)F$ where $F$ is the combustion flux and $\alpha$ the CO yield is how an inventory model would write this. If we linearise this relationship we can, I think, rewrite the problem with separate CO and $CO_2$ fluxes. With judicious choices of correlations we can probably ensure that the eigen-vectors of the prior covariance matrix (the space in which the inversion really takes place) reflect the underlying physical relationships. It's not obvious to me how to do this but the generated slopes between CO and $CO_2$ (as used in the paper) does seem a reasonable way to test it. So far so good.

The case for the observational correlations seems harder. What we shorthand as observational errors describe the differences between the modelled value with true inputs and the observed value (Rayner et al., 2019, Section 5). It encapsulates both model and instrumental/retrieval error. It is true that correlations in the observational covariance $\mathbf{R}$ do change the posterior uncertainty and mean for fluxes even if the fluxes are not coupled through the Jacobian.

Following notation of Rayner et al. (2019) consider the simplest case of two unknowns and two observations with an identity Jacobian $\mathbf{H} = \mathbf{I}$ and an identity prior covariance matrix $\mathbf{P} = \mathbf{I}$. Assume an observational covariance matrix $\mathbf{R} = \begin{smallmatrix} 1 & \alpha \\ \alpha & 1 \end{smallmatrix}$.

Applying the Sherman–Morrison–Woodbury formula (Cressie and Johannesson, 2008) to generate the posterior covariance $\mathbf{A}$ we see $\mathbf{A}_{1,1} = 1 - \frac{2}{4-\alpha^2}$. The key point here is that the posterior uncertainty, and consequently the posterior estimates for the fluxes are quite dependent on the choices of these correlation parameters. If this is true then the paper needs to spend some more effort on either justifying or testing the sensitivity to these parameters.

**Specific Comments**

**L447** Is an r value of 0.51 really moderate here, 25% of variance?

**L451** You should not be quoting p values here, the p value measures the chance that there is a relationship at all which is not interesting in this case.

**References**

Cressie, N. and Johannesson, G.: Fixed rank kriging for very large spatial data sets, Journal of the Royal Statistical Society: Series B (Statistical Methodology), 70, 209–226, 2008.

Palmer, P. I., Suntharalingam, P., Jones, D., Jacob, D. J., Streets, D. G., Fu, Q., Vay, S. A., and Sachse, G. W.: Using CO2: CO correlations to improve inverse analyses of carbon fluxes, Journal of Geophysical Research: Atmospheres, 111, doi:10.1029/2005JD006697, 2006.

Rayner, P. J., Michalak, A. M., and Chevallier, F.: Fundamentals of data assimilation applied to biogeochemistry, Atmospheric Chemistry and Physics, 19, 13 911–13 932, doi:10.5194/acp-19-13911-2019, https://www.atmos-chem-phys.net/19/13911/2019/, 2019.

---

## Author Comment (AC1) · 17 Feb 2021

We would like to express our gratitude to the anonymous reviewer for providing helpful comments. We certainly appreciate your time in reviewing this manuscript.

We apologize if this manuscript fails to convey succinctly our main points from the perspective of the reviewer. We will try to address each of the comments as best as we can. Certainly, we value your comments in improving this manuscript, as we also recognized the prospects of joint inversions in future studies.

————— Please see our responses to your comments below.

[Figure]

(General Comment). The manuscript "On the relationship between tropospheric CO and CO2 during KORUS-AQ and its role in constraining anthropogenic CO2" by Tang et al. presents analysis based on the CO, CO2 and 14CO2 data collected during the KORUS-AQ campaign over South Korea. It compares simulations of CO, CO2 and FFCO2 concentrations from a global transport model to the data and presents inversions of the FFCO2 emissions in East Asia (Eastern China, Korea and Japan) using this transport model and these data.

The prospect of the joint analysis of CO2, CO and 14CO2 data supported by transport model simulations and of their joint assimilation in an inversion system is very promising. Sometimes, the manuscript nearly reaches interesting insights on this topic. However, in a general way, the study and the manuscript fail to exploit the potential of such analysis. I think that more work and thoughts are needed to produce a paper that deserves publication and that this goes beyond what is usually done for "major revisions".

Response: Yes, this is a paper about the relationship between CO and CO2. We also emphasize that while our inverse analysis is the final key point of this work, the two other important key points that we are highlighting in this work are the following:

a) evaluation of CAM-Chem with CO2 and CO tags using prior fluxes from a posteriori fluxes from state-of-the-art global CO2 inversions (i.e., CAMSv117r1, CT2017, CTE2018). To our knowledge, this is the first development of a joint CO2 and CO modeling in community Earth system model, CESM/CAM-Chem. This is also one of the few studies in recent years that takes advantage of flux/source inversion products as priors in the model. We note that we have also compared our simulations to Carbon Tracker CO2 fields to ensure that our simulations are reasonable.

b) detailed description on CO and CO2 error statistics. Our intention is to highlight the need to evaluate the statistics of the CO and CO2 relationship using aircraft measurements, prior to any inverse analysis. We highlight, in particular, the need to provide spatiotemporal context on these relationships, which in previous inversion studies heavily rely on sensitivity tests. In this work, we highlight the importance of having knowledge of 'spatial error covariance localization length scales' prior to inversion.

———————— 1a) It is often very difficult to follow because the writing and the reasoning are not structured and rigorous enough. A large amount of sentences are confusing because of the lack of clarity, precision and explanations. The reading of the long series of statistics lacks of hierarchy. The use of simulations or datasets that are not much exploited in the analysis does not help. For instance, efforts are needed to follow, in section 3, whether or why under or over estimations of concentrations are supposed to highlight the under or over estimation of local, regional or global sources and sinks. The reading of section 4 is even more difficult.

Response: We are not quite clear as to what the reviewer is referring to about structure and hierarchy. As for Section 3, the aim is to evaluate the modeled abundance against aircraft measurements, which we summarized in Table 2. Our intention is to lay out the basis of error statistics as any typical model evaluation would conduct and provide some physical context on these errors. We recognized that these statistics (bias, correlation, rmse, as well as error correlation, enhancement ratio), while common, have direct relevance to Section 4 where we expound more on the spatial variability of these errors. We note that a more accurate representation of both the a priori error and observation error covariance matrices is in fact the crux of the inverse analysis, as with any other inversion methods that account for these uncertainties. We do emphasize the need to investigate the covariances and error covariances (and sensitivities) as these are the terms in the analysis that 'transfer information' to unobserved parameters that we are interested in estimating. We see this as an important step before any inverse analysis is undertaken.

The spatial data groupings correspond to different sampling regimes which help us understand issues of the CTM (i.e., representativeness, boundary layer, transport, mixing, emissions, sinks). Here, we provide a spatial and temporal context on the fidelity of the

model to capture these statistical measures, as these measures represent different components of the error characteristics.

Our emphasis is also to understand the vertical structure of these errors (Figure 4) and associate them to potential model deficiencies. As such, we used this section to compare not only with CarbonTracker (which we used for $CO_2$ initial condition) but also to CAMS, based on our previous work also published in this journal. As we mentioned above, we recognize the need to provide an evaluation of CAM-Chem to establish its fidelity (relative to state of the art models of $CO_2$) since this is the first development of such a system. We also note that this system is different from most models as we are taking advantage of 3 different a posterior fluxes (CAMS, CT2017, CTE2018) and used these as our priors. This is the reason why there is an added dimension to our model evaluation.

We approach this model evaluation similar to Tang et al. (2018) in order to be consistent with our previous work. We want to show across all the sections, the main essence of this paper: to show the relationship between CO and $CO_2$ and ffCO2. For our comparison with 14CO2 data, we note that to our knowledge this is the first publication of this data for this campaign with comparison to model equivalents. This is the reason why we have a separate sub-section. This is also very timely as there have been recent publications (also in this journal) about 14CO2 in nearby region.

It is unfortunate that the reviewer finds this section to be confusing. We do recognize that this manuscript is long but we chose to elaborate more on these statistics. This is the main reason why prior to our inverse analysis (Section 5), we expanded our discussion on error statistics to make sense of error covariances used in inverse analysis and to come up with reasonable 'localization' through the use of a subset of measurements closer to the sources (section 4.1 and 4.2 leading to section 4.3). We emphasize that this is not a manuscript of an inverse analysis of ffCO2 but more of elucidating the need to investigate the relationships prior to any inversions.

————— 1b) Furthermore, despite being relatively long, the text goes too fast on some of the critical parts of the reasoning like the rationale for the study in the introduction and the justification for the use of specific inversion configurations and parameters.

I think that many assumptions and analysis are debatable and that the manuscript shows a lack of hindsight on the topic and results of this study.

Response: We regret that the reviewer finds that many of our assumptions and analysis are debatable. We do recognize the length of this manuscript. As we mentioned above, we emphasis that this is not an inverse study per se. We intend to show the role of CO, CO2 and ffCO2 relationship on the inverse analysis (section 5) but only to briefly make an example. In this manuscript, we focus on discussing the spatial variability of the terms used in the inverse analysis and highlight the need to evaluate these relationships in CTMs. Again, the approach we have taken on this paper (i.e., revisiting Palmer et al. 2006) could open up discussion on the methodology for joint inverse analysis. Our intention is to bring to light some of the issues that need to be considered for future joint inversions, especially on the use of satellite retrievals for inversions, which lacks the vertical information that is critical in identifying transport-related errors that may confound any inverse analysis. This is not only true in single-species inversions but in fact is an added complication with multi-species inversions that should be carefully considered. There is relevance to adding CO observations in reducing uncertainties on top-down estimates of ffCO2; especially with the availability of atmospheric composition retrievals (CO, NO2) nowadays (i.e., TROPOMI, GOSAT-2). We emphasize the need to better understand the joint inversion, as this has yet to be investigated rigorously and that the species relationships (and associated errors) should not be taken for granted. As mentioned in the manuscript, our approach is based on widely used Bayesian synthesis inversion introduced by Enting (2002) and used by Baker et al. (2006), and more recently applied for multi-species inversion by Boschetti et al. (2018). While Wang et al. (2018) used a similar method (for single-species CO2 inversion), they focused on investigating the impact of reduced carbon component of CO2

based on CO on their estimates. We note that this approach, while with disadvantages, is still used in global inversions.

————— 2a) The specific scope and the objectives of sections 3 and 4 are not clear. This manuscript is the 5th one analyzing the CO2 and/or CO data from the KORUS-AQ campaign using transport models (after Tang et al. 2018, Halliday et al. 2019, Tang et al. 2019a and Gaubert et al. 2020, ACPD). How do section 3 and 4 draw on these previous publications and bring new learning?

Response: We thank the reviewer for pointing this out. As we mentioned above, this is the first evaluation of a community Earth system model (CESM/CAM-Chem) for joint CO2 and CO simulations. Tang et al. 2018 is about CAMS not CAM-Chem. We draw insights to this paper by comparing CAMS to CAM-Chem given the relatively different model configuration between these models. Halliday et al. 2019 is about enhancement ratios mostly focused on data. We draw insights to this paper by comparing their analysis to model equivalents especially with regards to the vertical distribution of enhancement ratios and correlations. As we mentioned in the Introduction, we view this as complementary (i.e., data and model comparison with inverse analysis). Tang et al. 2019a is about CO not CO2 using CAM-Chem. We draw insights on this work by using the optimized CO and error statistics (model vs data) which we used in our inverse analysis. Gaubert et al. 2020 is about data assimilation (with EnKF). We draw insights to this work by comparing tagging simulations that we did versus EnKF emission estimation that they conducted.

Note that this is the first time we provide model-data synthesis for CAM-Chem on both CO2 and CO. Here, we highlight the correlation (and error correlation) as this is really the crux of any inversions and clearly there is lack of detailed investigation reported in literature with regards to these covariances for joint inversions. And if we are to proceed with joint global inversions, we ought to be aware of the deficiencies of the CTMs in capturing these relationships. And what better way to do this but by comparison with aircraft data. Also, note that 14CO2 during KORUS-AQ is first reported in this

manuscript. If there's new learning on this manuscript, it would be the synthesis of all these in the context of inverse analysis.

————— 2b) Furthermore, opposed to what is claimed repeatedly (e.g. in the title, the abstract, the beginning of section 1.1, the beginning of the conclusion...), I hardly see how hese sections feed the configuration (in particular the error covariance matrices) or the analysis of the inversions in section 5.

The introduction does not help much. Until the beginning of section 1.1, this is a collection of very general (and sometimes misleading) statements about the monitoring CO2 anthropogenic emissions using atmospheric data. Section 1.1 fails to bring clear specific context, rationale and objectives for this study. Lines 150-154 look like a summary of the activities rather than a list of objectives. Lines 154-172 attempt to distinguish this new study from the previous ones by pointing to practical differences that are hardly convincing. Lines 769-770 and 775-776 claim that the set-up of matrix Se follows values of errors on CO and CO2 and errRCO,CO2 from section 4 but I do not see how. Anyhow, the errors and the correlation of errors in CO and CO2 modeled concentrations are driven by the atmospheric transport, the surface fluxes (and other source and sinks), and to errors in both the transport model and in the modeled sources and sinks. I do not see why it should be used to characterize transport model errors only. The authors seem to miss the links between their statistics of model-data differences and Sa (see below my general comment about the key role that this matrix should have played).

Response: We thank the reviewer for what appears to be a thorough read of our manuscript. We value your comments. We again regret that the reviewer fails to find a clear understanding of the flow. This may be due to the length of the manuscript. It is quite unfortunate that the reviewer views the Introduction as a "collection of very general (and sometimes misleading) statements". Our intent was to provide context on the role of CO to GHG and ffCO2 in particular. Why CO? We apologize if our interpretation of literature is inaccurate from the reviewer's perspective. We would appreciate it though if the reviewer points out some of these inaccuracies and misleading statements.

To reiterate, Section 3 is about comparison with data and models. Section 4 is about error correlation and enhancement ratios. Section 5 is about a simple demonstration of a joint inverse analysis. Section 4 feeds to Section 5 through the analysis of the spatial extent of these errors (especially in the vertical). We use the error correlations from Section 4 in particular and localized the dataset we used in the inversion to account for potential confounding factors (transport errors) especially aloft. Again, this paper is not all about joint inverse analysis.

We would like to repeat our response to Reviewer 2 with regards to Sa and Se as this is also relevant in later comments of the reviewer:

Palmer et al., (2006), as well as Boschetti et al. (2018), relied on tagging CO2 and CO components using a CTM (linearizing both CO2 and CO) to conduct a suite of Bayesian synthesis inversions. Boschetti et al. (2018), however, utilized STILT to obtain these Jacobians.

Within the CTM (where both CO2 and CO are simulated), there is, however, a physically-based coupling between these runs through the use of emission ratios in prior emissions for ffCO2 and CO. That is, they are both emitted to the atmosphere in the model by sector-specific assumptions of emission factors per species, and associated emission ratio between species. As we showed in Tang et al. (2019b), for a particular combustion process, we can estimate the combustion products by stoichiometry:

— please see figure 1 for equations and figures —

And so, if we are to look at the equivalent atmosphere abundance of these emission tags (as they are emitted to the atmosphere), they are mostly correlated near the sources, with correlations diminishing downwind of the source. Obviously, observing the relationship downwind does not imply that the correlations (or the lack thereof) are created by this co-emission alone, but typically also through mixing, transport, and associated sinks. These are further quantified in the inverse analysis through the a priori error covariance matrix (i.e., errors in emission ratios) and observation error covariance matrix (i.e., errors in transport, mixing, among others).

We note here that a more accurate representation of both the a priori error and observation error covariance matrices is in fact the crux of the analysis, as with any other inversion methods that account for these uncertainties. We do emphasize the need to investigate the covariances and error covariances (and sensitivities) as these are the terms in the analysis that 'transfer information' to unobserved parameters that we are interested in estimating.

We agree that it is worth comparing with what happens in nature. In fact, this is one of our key points – i.e., can we evaluate this 'modeled' relationship? Are these relationships that can be derived from observations of the species abundance in nature reasonably captured by CTMs? We agree with Peter regarding the expression on sources. Please see our response to Comment 3). In essence, we are looking at $\alpha$ ( effective emission factor in our notation), which we argue should be consistently represented in the a priori inventory for both CO and CO2. At present, with few exceptions, ffCO2 is derived from energy statistics separate from air pollution emission inventories.

We note, however, that as we previously mentioned the yield is quantified as effective emission factor, and the emission ratio is the quantity we can evaluate with observations by looking at the derived enhancement ratio ($\Delta$CO/$\Delta$CO2) from observations of CO and CO2 abundance. Note that this can only be 'appropriately' evaluated with observations near the emission sources, just like studies using field campaign measurements to estimate emission ratio. Otherwise, the enhancement ratio is confounded by mixing and transport and hence cannot be taken and interpreted as the abundance equivalent of emission ratio. See for example Mauzerall et al. (1998) Figure 4 (shown below):

[Figure]

— please see figure 2 for equations and figures —

where, the regression line of [X] and [Y] (i.e., $(\Delta[Y])/(\Delta[X])$) downwind is steeper than over a fresh plume. This complication arises from various sources and sinks, other than combustion which in reality are ubiquitous. And so, here we take the enhancement ratio only as a key diagnostic on the bulk relationship between CO and CO2 (resulting from a combination of emission ratio, mixing and sinks).

Palmer et al. (2006) investigated the sensitivities of different assumptions of activity, and emission factors and concluded that errors in emission factors largely contribute to the uncertainty estimate. Rather than 'reinventing the wheel', we took a more conservative value of 0.5 (Boschetti et al. 2018 used 0.7, while Palmer et al. 2006 suggested >0.7). As we noted in the manuscript, there is little information from flux measurements to verify this error correlation. Also, note that these correlations, when used in the inversion, is not correlation between CO and CO2 abundance near the source but the correlation of the errors between modeled CO and CO2 abundance.

We agree on the proper interpretation of observation error covariances, R. We also agree on the influence of R on the Bayesian synthesis inversion mean estimate and a posteriori error covariance estimate. In fact, both P (or B) and R influence the first and second moment estimates of the conditional pdf. This is the main reason why prior to our inverse analysis (section 5), we expanded our discussion on error statistics to make sense of R and to come up with reasonable localization of R through the use of a subset of measurements closer to the sources (section 4.1 and 4.2 leading to section 4.3).

This influence is clearly more complex but it shows as well that the a posteriori error covariance is a function of both the a priori error and observation error covariances. While the Jacobians (tags) itself should exhibit some correlation as the species are co-emitted, we need to quantify the uncertainties, not only on error variances but more importantly on the error correlations.

We note as well that our inverse analysis is a product of this investigation on R. By way of expressing the sensitivity (enhancement ratio) with correlation (see related response in Comment 3), we get:

— please see figure 3 for equations and figures —

The same relationship holds for the error sensitivity. In this manuscript, we focus on discussing the spatial variability of these terms, as well as the corresponding magnitudes, so as to guide us in defining the cross-species error correlation components of R (see Eq. 4 of our manuscript). In fact, we used a more conservative error correlation of 0.33 (Boschetti et al. 2018 and Palmer et al. 2006 used 0.7). We opted not to show our tests where we vary our assumptions of these values as this was already done by Palmer et al. (2006). Rather, we choose to emphasize the need to have an estimate of these values based on data, especially aircraft measurements given known issues with vertical transport/mixing in CTMs. We intend to show the role of this relationship on the inverse analysis (section 5) but only to briefly make an example. We emphasize that this is not a manuscript of an inverse analysis of ffCO2 but more of elucidating the need to investigate the relationships prior to any inversions.

As an aside, if we are to recast the first moment of p(x|y):

— please see figure 4 for equations and figures —

————— 3a) There is a critical lack of proper discussion on the spatial extent and scales that are suitable for the analysis of the data. Parts of sections 3 to 5 attempt to distinguish the influence of Eastern Asia or the rest of the world vs that of Korea, l340-350 discuss the sources overflown during the campaign and the flights over the West Sea are said to be designed to capture "China pollution outflow". Some sentences even raise (too late) some concerns associated to the coarse spatial resolution of CAM-chem. However, in general, and in particular in the title, introduction and section 2, there is no real reasoning regarding the observation footprints and regarding the modeling and inversion domain and resolution that are suited for these data

Response: This is quite unfortunate as this is one of the main theme of this manuscript. We revisited Palmer et al. (2006) which uses a global CTM and aircraft data over similar region. We are well cognizant of the representativeness of these measurements. This is why we ended up aggregating our tags (emission regions). In particular, we only focused on East Asia and Korea+Japan. This is also the reason why we use more local data close to the source rather than using all the data. We note however that we have compared to CarbonTracker and CAMS of similar resolution. Again, the inverse analysis is intended to show an example as knowledge of enhancement ratio and correlation extends beyond inverse analysis.

————— 3b) The introduction mixes all inversion scales and all types of observation networks. Nothing is said about the wind fields during the campaign. The comparisons of a single global coarse resolution model to TCCON individual sites, to OCO-2, to MOPITT data or to the aircraft data are brought together without consideration for the differences between these observation datasets in terms of flux and process representativeness. The statistics obtained here are sometimes compared with results from other campaigns in different regions or with different coverage, or from "state-of-the-art" models (without mentioning whether they are global or regional). The contrast between the spatial extent of the regions for which the total emissions are rescaled by the inversion and the local (nearly vertical) reading of the data footprints at l340-350 is questioning.

Given the spatial extent of the KORUS-AQ campaign (less than 8âŮęx8âŮę) and its density of data over South Korea, the use of a global 1âŮę resolution transport model with coarse vertical resolution to analyze it is not obvious and sounds like a step backward compared to the previous publications on this campaign (which used higher resolution models). From what I understand, the model is interpolated at each observation location and all the statistics are derived over the ensemble of observation locations. What can be the meaning of such statistics at a resolution much finer than that of the model?

Response: We compared our results to CarbonTracker as well. CAMS is also compared with CAM-Chem. Palmer et al. (2006) used a global CTM. There are several global inversions as well in literature that in fact used regional datasets. The intent of using TCCON, OCO-2 and MOPITT is to provide a comprehensive (albeit limited in time) evaluation on the extent of CAM-Chem (relative to other models) to capture not just the regional scale but global features of both CO and $CO_2$ (e.g., surface and column, synoptic and boundary layer). There's obvious (and understandable) argument about using global CTMs to estimate regional sources. But we have carefully presented all these statistics and show that at some of these scales, it is appropriate to estimate regional sources. Our evaluation of $CO_2$ is comparable to CarbonTracker and CAMS. Having said this, we will modify our manuscript to ensure we mention the appropriate scales of these comparisons.

It is unclear as to what the reviewer is referring to with regards to the statement "brought together without consideration for the differences between these observation datasets in terms of flux and process representativeness". Across more than 20 years of inversion literature, we are well-cognizant of the issue of representativeness especially with global models. We will try to revise this manuscript to provide more emphasis. However, providing proofs and arguments related to representativeness and aggregation errors is beyond the scope for this manuscript.

We would like to reiterate here that joint inversions are relatively new. While dealing with confounding factors is necessary, we intend to show that CTMs that do not capture even the large-scale features of the CO and $CO_2$ relationship would have difficulty taking advantage of the information that CO brings. In fact, we emphasize that joint inversion of CO and $CO_2$ has to be conducted at scales commensurate with the lifetime of CO. Our presentation of the statistics is an honest way to show the scales at which CAM-Chem has captured.

————— 4a) The usual concept for the co-assimilation of CO and $CO_2$ is that the signal from misfits between modeled and measured CO can be used to add constraint

on the inversion of FFCO2 emissions because uncertainties in the FFCO2 emissions are connected to uncertainties in FFCO emissions. This usually translates into two options for the inversion: (a) rescaling activity levels underlying both FFCO and FFCO2 emissions rather than FFCO and FFCO2 emissions separately or (b) rescaling separately FFCO and FFCO2 emissions accounting for positive correlations between their respective prior uncertainties. The key challenge for joint CO-CO2 inversions is usually thought to be the uncertainties and high spatial and temporal variations in the CO/CO2 emission ratios i.e. in (a) the uncertainties and variations in the CO and CO2 emission factors to be multiplied by the common activity indices to get emissions and in (b) the level and variations of the positive correlation between the prior uncertainties in FFCO and FFCO2 emissions. The authors of this study control separately FFCO and FFCO2 emissions but neglect the correlations between their respective prior uncertainties or acknowledge that they have no idea about how to parameterize them, cutting the critical connection between FFCO2 and FFCO (see lines 819-830). The configuration of Sa is actually made all the more difficult by using two different inventories for prior FFCO2 and FFCO emissions, which undermines the ability to rely on tights connections between these emissions (especially since the CO inventory has been multiplied by 2 to better fit the data before the inversion).

Response: Please see our response to comment 2b) and to Reviewer 2 on Sa. While these inventories are taken from different sources, we argue that evaluating the bulk enhancement ratios would provide insights on their inconsistencies. We highlight in fact the need to have consistent emission inventories for GHG and air quality. This manuscript is trying to revisit Palmer et al. (2006) and open up discussion on meaningful ways on how these correlative measurements can be used effectively in ffCO2 estimation.

——————— 4b) The authors even assume that these correlations are negative, i.e. that the combustion efficiency could be the main source of uncertainty in both FFCO2 and FFCO emissions. However, the prior FFCO and FFCO2 emissions are based on different inventories (and FFCO emissions have been multiplied by 2), so that, in principle, the ratios between these emissions should not correspond to an assumption on this efficiency. In a more general way, such an assumption is quite surprising. Uncertainties in the level of activity at national, and even more at regional to local scales should be a dominant source of uncertainty in FFCO2 emission inventories (especially at sub-annual temporal scales). Such a driver of uncertainty in FFCO2 emissions raises correlations with uncertainties in FFCO emissions. Even if uncertainties in CO emission factors are one of the main sources of uncertainties in FFCO emissions, their counterpart in FFCO2 emissions (the generation of plus or minus CO2 depending on the combustion efficiency) can hardly balance this driver. The assumption of negative correlations between uncertainties in FFCO2 and FFCO prior emissions is further weakened by various discussions in sections 3 and 4 that point to common underestimation or overestimation of sources in both the CO2 and CO inventories.

Response: Please see our response to comment 2b) and to Reviewer 2 on Sa as well. Based on Palmer et al. (2006), they found that the emission factor contributes largely to the uncertainty. Looking at emission ratios, which to an extent (Eq A3) can be proportional to enhancement ratio especially near the source, activity cancels out. The utility of CO lies not on the magnitude alone but its sensitivity to CO2 (i.e., its relationship). Please see our recasting of the a posteriori estimates in our response to comment 2b as well. We agree with the reviewer that it is non-trivial to attribute the mismatch on specific sectors or to activity alone. We cannot differentiate this with the data so far nor can we differentiate this with CO alone. Our intention here is to highlight the information content of CO on ffCO2 estimation. But in order to do this, we need to ensure that we understand the error statistics and that the CTMs capture this relationship. Again, the inverse analysis is an example of showing this information content.

———— 4c) The idea that Se could ensure the expected connection between FFCO2 and FFCO is not relevant. Se prevents the inversion from overfitting the data and limits

the corrections to the prior emissions. Correlations between CO2 and CO transport modeling errors help the inversion better filter such errors when deriving FFCO2 and FFCO emisions but this has a very indirect and weak impact on the connection between these emissions. One could think that the results from the inversions in section 5 contradict these statements. However, I assume that the consistency between FFCO2 and CO2- CO inversions and their divergence from CO2 inversions is intrinsically linked to the very low dimension of these inversions (with 4 or 8 unknowns). Various assumptions and settings of these inversion raise issues (see below) and I thus have the feeling that this result is not robust.

The general topic of the CO-CO2 correlations that should be at the heart of this paper is complex and I feel that the authors missed it.

Response: We beg to disagree. As with Palmer et al. (2006), in the Bayesian synthesis inversion framework, Se does not only represent observation error covariances, but more dominantly represent (supposedly) model errors including transport, mixing and sinks. Hence, the mismatch (innovation) reflects errors in capturing the relationship – perhaps not on the emission level but on the representation of the variability of the enhancement ratio due to differences in lifetime of the species (e.g., sources and sinks). The main issue with these inversions is the fact that we treat the model as perfect and it is extremely difficult to quantify model errors. Errors on the Jacobian (which we are trying to assess) should not be taken for granted.

Yes, the consistency is due to lower dimension; mainly as a result of our effort to understand the scales at which the information content of CO can be utilized effectively.

It is again unfortunate that the reviewer sees the general essence of this work as something that we missed. We will continue on this work and refine our statements to improve on clarity.

We note that this concept, while old, is already difficult to grasp. How much more by introducing this to inverse analysis. We emphasize that there are only very few studies

about joint inversion, mainly due to the misconception that the inherent variability of these relationships is far greater than the information content. We argue that we need to start thinking about extracting information more effectively.

———————— 5a) I don't understand how global inversions (even if they focus on East Asia) assimilating CO2 (with or without CO) can behave correctly when inverting scaling factors for FFCO2 emissions only and keeping other CO2 fluxes fixed. The "posterior" biogenic fluxes from CTE and CAMS bear large uncertainties. The problem is enhanced by the lack of spatial resolution of the inversion, which does not help distinguish the patterns from anthropogenic vs biogenic fluxes in the data.

Response: This is a valid point. We do not have a concrete answer to this. We view this as a signal-to-noise issue: i.e., we should see a large source of ffCO2 over East Asia and that expect to be able to 'track' this with CO. We also think that posterior errors are smaller than prior errors especially for CESM/CAM-Chem, which is an Earth system model with dynamic vegetation. We view this as well as a two-step approach (or iterative), where the expectation is that posterior fluxes fit atmospheric abundance better than the prior fluxes (but obviously perhaps for the wrong reason).

———————— 5b) I do not understand the rationale behind removing observations above 3 km (the "localization purposes": l791-792 refer to "previous section" but I do not see where sections 3 and 4 help understand this choice). The inversions are global, and data above 3 km should be helpful for constraining ROW.

Response: We selected the data where ffCO2 signal is relatively strong (Figure 8 of Section 4). In essence, we view this as reducing the noise. For example, data above 3km do not really have ffCO2 signal from both East Asia and Korea. You can also think about this as tapering the radius of influence of a given datapoint by localizing its impact – similar to error covariance localization in EnKF. Here, we are effectively localizing the Jacobians and use of data to lower than 3km.

———————— 5c) I have some doubts regarding the way the FFCO2 simulations are compared to gradients of FFCO2 from gradients in 14CO2 observations: I believe that the authors should have computed, in the FFCO2 simulations, the gradients to the location of the 14CO2 background sites rather than introduce a "ffCO2 offset".

Response: This is a valid point. We think that our global ffCO2 is under-estimated and that spatial gradient (variability) is ffCO2 offset is small. We view this to be similar to a constant in a regression analysis. However, we acknowledge this issue. Thank you for the suggestion.

————— 5d) There is a lack of consideration for the inventories. As mentioned above, the fact that the FFCO2 and FFCO inventories are quite independent strongly hampers the interpretation of the model-data comparisons and the reading of CO/CO2 ratios in the model. The EDGAR inventory, which plays a critical role in this study (this is the FFCO2 emission inventory behind the simulations in sections 3 and 4 and the prior emission inventory in section 5) is not even named in the main text, the authors speaking about CTE and CAMS fluxes (which combine natural fluxes optimized through global inversions with the EDGAR inventory) only. Differences between the CAMS analysis and CAM-Chem simulations are not supported by insights on the emission maps behind these two simulations. The FFCO2 emissions from the inversion are hardly compared to official inventories (and the CO emissions from the inversions are not discussed).

Response: We mentioned it in Table 1 and Table S1. We showed a regional budget for ffCO2 in Table S2. In addition, we showed the spatial distribution of the fluxes in Figure S1.

We do acknowledge that we did not choose to compare rigorously with emission inventories at this stage of our work, given that we are only starting to get a handle of this relationship and full, comprehensive inversion is beyond the scope of this work at this point.

————— 5e) I could draw a long list of secondary issues in specific paragraphs or

sentences (and even in the mathematical notations and equations) but I restrain myself to the main ones raised above

Response: Your input is always welcome. We follow the notations by Rodgers 2001 and Palmer et al. 2006. Similar notations are used by Boschetti et al. (2018) as well.

——————— 5d) In conclusion, I think that the scope and configuration of the analysis should be rethought, and that the presentation of such a study should be strongly improved. I thus recommend this manuscript to be rejected and re-submitted after a deep revision.

Response: Thank you for your comments. We will revisit the manner in which we presented our study for this manuscript.

————————

References (not cited in the manuscript):

Mauzerall, D.L., Logan, J.A., Jacob, D.J., Anderson, B.E., Blake, D.R., Bradshaw, J.D., Heikes, B., Sachse, G.W., Singh, H. and Talbot, B., 1998. Photochemistry in biomass burning plumes and implications for tropospheric ozone over the tropical South Atlantic. Journal of Geophysical Research: Atmospheres, 103(D7), pp.8401-8423.

$$C_{x_1}H_{x_2}O_{x_3}N_{x_4}S_{x_5} + n_1(1+e)(O_2 + 3.76N_2) \rightarrow$$
$$n_2CO_2 + n_3H_2O + n_4O_2 + n_5N_2 + n_6CO + n_7NO + n_8NO_2 + n_9SO_2 + n_{10}C + \cdots \quad \text{Eq. (A1)}$$

where emissions of these intermediate products are typically expressed as:

$$E_x = \sum_s [A_s \cdot EF_{x,s} \cdot (1 - CE_{x,s})]$$
$$= \sum_s [A_s \cdot EEF_{x,s}] \quad \text{Eq. (A2)}$$

Here, $E_x$ is the total mass of emissions for species $x$, $EF_{x,s}$ is its associated emission factor for a specific source/sector $s$, and $A_s$ is the activity level of the source. $CE_{x,s}$ corresponds to effectiveness of control measure and $EEF_{x,s} = EF_{x,s} \cdot (1 - CE_{x,s})$ is the effective emission factor. When we take the ratio of emissions (Eq. A2) of co-emitted species $x$ and $y$, we obtain

$$\frac{E_y}{E_x} = \frac{\sum_s [A_s \cdot EEF_{y,s}]}{\sum_s [A_s \cdot EEF_{x,s}]}$$
$$= \sum_s \left(\frac{EEF_{y,s}}{EEF_{x,s}}\right)\left(\frac{E_{x,s}}{E_{x,total}}\right) \quad \text{Eq. (A3)}$$

This ratio can be expressed as the sum of the products of the ratio of effective emission factors ($R_{x,y,s}^{EEF}$) and the fractional contribution of emission sector f for species ($f_{x,s}$).

**Fig. 1.**

[Figure]

**Figure 4a.** Schematic of plume formation and transport. The $(x_p, y_p)$ and $(x_b, y_b)$ indicate mixing ratios within the fresh plume at the time of sampling and in background air in proximity to the fire, respectively. The $(x_p', y_p')$, and $(x_b', y_b')$ indicate mixing ratios within the plume sampled by the aircraft and in background air in the vicinity of the sampled plume, respectively.

[Figure]

**Figure 4b.** Schematic of plume sampling. Long-dashed line indicates a fresh plume with enhancements $(x_p-x_b, y_p-y_b)$ relative to background mixing ratios near the fire $(x_b, y_b)$. Dotted line indicates a plume remote from the fire which the aircraft measures relative to local background mixing ratios $(x_b', y_b')$. Enhancement ratios are obtained from the slopes of these lines.

**Fig. 2.**

$$\frac{\Delta CO}{\Delta CO_2} = \frac{cov(CO, CO_2)}{var(CO_2)} = \rho_{CO,CO_2} \frac{\sigma_{CO}}{\sigma_{CO_2}}$$

**Fig. 3.**

$$x^a = (H^T R^{-1} H + B^{-1})^{-1}(H^T R^{-1} H y + B^{-1} x^b)$$

or

$$x^a = x^b + B H^T (H B H^T + R)^{-1}(y - H x^b)$$

to:

$$x^a - x^b = B H^T (H B H^T)^{-1}(H B H^T)(H B H^T + R)^{-1}(y - H x^b)$$

we can easily recognize that the increment $x^a - x^b$ is a function of three terms:

1) Innovation: $(y - H x^b)$, which is the mismatch of observed and modeled abundance
2) Least squares fit $(H B H^T)(H B H^T + R)^{-1}$ which is the ratio of the error variances
3) Linear regression: $B H^T (H B H^T)^{-1} = cov(H e_x, e_y)/var(e_y)$ where $B = cov(e_y, e_y{}^T)$ which is the slope of a line.

In this manuscript, we highlight the need to investigate 3).

**Fig. 4.**

[Figure]

---

## Author Comment (AC2) · 17 Feb 2021

We thank Peter Rayner for his very helpful comments. We certainly appreciate your time in reviewing this manuscript. Most importantly, we value your comments in improving this manuscript.

We do recognize that the approach we have taken on this paper (i.e., revisiting Palmer et al. 2006) could open up discussion on the methodology for joint inverse analysis. Our intention is to bring to light some of the issues that need to be considered for future joint inversions, especially on the use of satellite retrievals for inversions, which lacks the vertical information that is critical in identifying transport-related errors that may

confound any inverse analysis. This is not only true in single-species inversions but in fact is an added complication with multi-species inversions that should be carefully considered.

————- Please see our responses to your comments below.

1) this paper examines the role of carbon monoxide (CO) measurements in a joint inversion of CO and CO2 fluxes over East Asia. The paper builds on methodologies and results from some previous papers but delves further into the specific role of CO in improving the inversion.

Response: Yes, this is a paper about the relationship between CO and CO2. We also emphasize that while our inverse analysis is the final key point of this work, the two other important key points that we are highlighting in this work are the following:

a) evaluation of CAM-Chem with CO2 and CO tags using prior fluxes from a posteriori fluxes from state-of-the-art global CO2 inversions (i.e., CAMSv17r1, CT2017, CTE2018). To our knowledge, this is the first development of a joint CO2 and CO modeling in community Earth system model, CESM/CAM-Chem. This is also one of the few studies in recent years that takes advantage of flux/source inversion products as priors in the model. We note that we have also compared our simulations to Carbon Tracker CO2 fields to ensure that our simulations are reasonable.

b) detailed description on CO and CO2 error statistics. Our intention is to highlight the need to evaluate the statistics of the CO and CO2 relationship using aircraft measurements, prior to any inverse analysis. We highlight, in particular, the need to provide spatiotemporal context on these relationships, which in previous inversion studies heavily rely on sensitivity tests. In this work, we highlight the importance of having knowledge of 'spatial error covariance localization length scales' prior to inversion.

————- 2) I found the paper difficult to assess mainly because I have never felt comfortable that I understand the methodology of Palmer et al. (2006). It's quite possible

therefore that what I'm about to say is wrong. I also want to add that I am taking advantage of the discussion period to comment on the paper, i.e. this is not a review.

Response: We recognize the potential difficulty. As mentioned, we intend to open up the discussion on such topic given the relevance of adding CO observations in reducing uncertainties on top-down estimates of ffCO2; especially with the availability of atmospheric composition (reactive gases) retrievals (CO, NO2) nowadays (i.e., TROPOMI, GOSAT-2). CO is also used mention that CO is used for BBCO2, and usually as a separate step (Liu et al., Science).We emphasize the need to better understand the joint inversion, as this has yet to be investigated rigorously and that the species relationships (and associated errors) should not be taken for granted. As mentioned in the manuscript, our approach is based on widely used Bayesian synthesis inversion introduced by Enting (2002) and used by Baker et al. (2006), and more recently applied for multi-species inversion by Boschetti et al. (2018). While Wang et al. (2018) used a similar method (for single-species CO2 inversion), they focused on investigating the impact of reduced carbon component of CO2 based on CO on their estimates.

———- 3) As I understand it the Palmer et al. (2006) methodology defines independent Jacobians for CO and CO2 i.e., there is no physical coupling between the two sources or between each source type and the other observation type. These relationships are introduced by correlations in the observation and prior covariance matrices. It's obvious this works for prior correlations and less obvious (though still true) that it works for observations.

Response: Yes, Palmer et al., (2006), as well as Boschetti et al. (2018), relied on tagging CO2 and CO components using a CTM (linearizing both CO2 and CO) to conduct a suite of Bayesian synthesis inversions. Boschetti et al. (2018), however, utilized STILT to obtain these Jacobians.

Within the CTM (where both CO2 and CO are simulated), there is, however, a physically-based coupling between these runs through the use of emission ratios in

prior emissions for ffCO2 and CO. That is, they are both emitted to the atmosphere in the model by sector-specific assumptions of emission factors per species, and associated emission ratio between species. As we showed in Tang et al. (2019b), for a particular combustion process, we can estimate the combustion products by stoichiometry:

–please see figure 1 for equations and figures–

And so, if we are to look at the equivalent atmosphere abundance of these emission tags (as they are emitted to the atmosphere), they are mostly correlated near the sources, with correlations diminishing downwind of the source. Obviously, observing the relationship downwind does not imply that the correlations (or the lack thereof) are created by this co-emission alone, but typically also through mixing, transport, and associated sinks. These are further quantified in the inverse analysis through the a priori error covariance matrix (i.e., errors in emission ratios) and observation error covariance matrix (i.e., errors in transport, mixing, among others).

We note here that a more accurate representation of both the a priori error and observation error covariance matrices is in fact the crux of the analysis, as with any other inversion methods that account for these uncertainties. We do emphasize the need to investigate the covariances and error covariances (and sensitivities) as these are the terms in the analysis that 'transfer information' to unobserved parameters that we are interested in estimating.

——— 4). It is worth comparing the approach with what happens in nature. Sources are clearly related since some CO2 fluxes are the result of combustion which may also produce CO with an uncertain yield. Writing the problem as FCO = $\alpha$F, FCO2 = $(1-\alpha)$F where F is the combustion flux and $\alpha$ the CO yield is how an inventory model would write this. If we linearise this relationship we can, I think, rewrite the problem with separate CO and CO2 fluxes. With judicious choices of correlations we can probably ensure that the eigen-vectors of the prior covariance matrix (the space in

which the inversion really takes place) reflect the underlying physical relationships. It's not obvious to me how to do this but the generated slopes between CO and CO2 (as used in the paper) does seem a reasonable way to test it. So far so good.

Response: We agree that it is worth comparing with what happens in nature. In fact, this is one of our key points – i.e., can we evaluate this 'modeled' relationship? Are these relationships that can be derived from observations of the species abundance in nature reasonably captured by CTMs? We agree with Peter regarding the expression on sources. Please see our response to Comment 3). In essence, we are looking at $\alpha$ ( effective emission factor in our notation), which we argue should be consistently represented in the a priori inventory for both CO and CO2. At present, with few exceptions, ffCO2 is derived from energy statistics separate from air pollution emission inventories.

We note, however, that as we previously mentioned the yield is quantified as effective emission factor, and the emission ratio is the quantity we can evaluate with observations by looking at the derived enhancement ratio ($\Delta$CO/($\Delta$ãĂŰCOãĂŮ_2 )) from observations of CO and CO2 abundance. Note that this can only be 'appropriately' evaluated with observations near the emission sources, just like studies using field campaign measurements to estimate emission ratio. Otherwise, the enhancement ratio is confounded by mixing and transport and hence cannot be taken and interpreted as the abundance equivalent of emission ratio. See for example Mauzerall et al. (1998) Figure 4 (shown below):

–please see figure 2 figures–

where, the regression line of [X] and [Y] (i.e., ($\Delta$[Y])/($\Delta$[X])) downwind is steeper than over a fresh plume. This complication arises from various sources and sinks, other than combustion which in reality are ubiquitous. And so, here we take the enhancement ratio only as a key diagnostic on the bulk relationship between CO and CO2 (resulting from a combination of emission ratio, mixing and sinks).

Palmer et al. (2006) investigated the sensitivities of different assumptions of activity,

and emission factors and concluded that errors in emission factors largely contribute to the uncertainty estimate. Rather than 'reinventing the wheel', we took a more conservative value of 0.5 (Boschetti et al. 2018 used 0.7, while Palmer et al. 2006 suggested >0.7). As we noted in the manuscript, there is little information from flux measurements to verify this error correlation. Also, note that these correlations, when used in the inversion, is not correlation between CO and CO2 abundance near the source but the correlation of the errors between modeled CO and CO2 abundance.

——— 5) The case for the observational correlations seems harder. What we shorthand as observational errors describe the differences between the modelled value with true inputs and the observed value (Rayner et al., 2019, Section 5). It encapsulates both model and instrumental/retrieval error. It is true that correlations in the observational covariance R do change the posterior uncertainty and mean for fluxes even if the fluxes are not coupled through the Jacobian.

Following notation of Rayner et al. (2019) consider the simplest case of two unknowns and two observations with an identity Jacobian H = I and an identity prior covariance matrix P = I. Assume an observational covariance matrix R = 1 $\alpha$ $\alpha$ 1.

Applying the Sherman–Morrison–Woodbury formula (Cressie and Johannesson, 2008) to generate the posterior covariance A we see $A\_{1,1} = 1-2/(4-\alpha^2)$. The key point here is that the posterior uncertainty, and consequently the posterior estimates for the fluxes are quite dependent on the choices of these correlation parameters. If this is true then the paper needs to spend some more effort on either justifying or testing the sensitivity to these parameters.

Response: We agree with Peter on the proper interpretation of observation error covariances, R. We also agree on the influence of R on the Bayesian synthesis inversion mean estimate and a posteriori error covariance estimate. In fact, both P (or B) and R influence the first and second moment estimates of the conditional pdf. This is the main reason why prior to our inverse analysis (section 5), we expanded our discussion

on error statistics to make sense of R and to come up with reasonable localization of R through the use of a subset of measurements closer to the sources (section 4.1 and 4.2 leading to section 4.3).

To show a similar derivation by Peter for two observations, if we let:

–please see figure 3 for equations–

This is clearly more complex but it shows as well that the a posteriori error covariance is a function of both the a priori error and observation error covariances. While the Jacobians (tags) itself should exhibit some correlation as the species are co-emitted, we need to quantify the uncertainties, not only on error variances but more importantly on the error correlations.

We note as well that our inverse analysis is a product of this investigation on R. By way of expressing the sensitivity (enhancement ratio) with correlation (see related response in Comment 3), we get:

–please see figure 4 for equations–

The same relationship holds for the error sensitivity. In this manuscript, we focus on discussing the spatial variability of these terms, as well as the corresponding magnitudes, so as to guide us in defining the cross-species error correlation components of R (see Eq. 4 of our manuscript). In fact, we used a more conservative error correlation of 0.33 (Boschetti et al. 2018 and Palmer et al. 2006 used 0.7). We opted not to show our tests where we vary our assumptions of these values as this was already done by Palmer et al. (2006). Rather, we choose to emphasize the need to have an estimate of these values based on data, especially aircraft measurements given known issues with vertical transport/mixing in CTMs. We intend to show the role of this relationship on the inverse analysis (section 5) but only to briefly make an example. We emphasize that this is not a manuscript of an inverse analysis of ffCO2 but more of elucidating the need to investigate the relationships prior to any inversions.

As an aside, if we are to recast the first moment of p(x|y):

–please see figure 5 for equations–

――― Specific Comments:

1) L447 Is an r value of 0.51 really moderate here, 25% of variance?

Response: Thank you. We should change it to 'fair'.

―――

2) L451 You should not be quoting p values here, the p value measures the chance that there is a relationship at all which is not interesting in this case.

Response: Thank you. We should not quote p values for this.

――― References (not cited in the manuscript):

Mauzerall, D.L., Logan, J.A., Jacob, D.J., Anderson, B.E., Blake, D.R., Bradshaw, J.D., Heikes, B., Sachse, G.W., Singh, H. and Talbot, B., 1998. Photochemistry in biomass burning plumes and implications for tropospheric ozone over the tropical South Atlantic. Journal of Geophysical Research: Atmospheres, 103(D7), pp.8401-8423.

[Figure]

$$C_{x_1}H_{x_2}O_{x_3}N_{x_4}S_{x_5} + n_1(1+e)(O_2 + 3.76N_2) \rightarrow$$
$$n_2CO_2 + n_3H_2O + n_4O_2 + n_5N_2 + n_6CO + n_7NO + n_8NO_2 + n_9SO_2 + n_{10}C + \cdots \quad \text{Eq. (A1)}$$

where emissions of these intermediate products are typically expressed as:

$$E_x = \sum_s [A_s \cdot EF_{x,s} \cdot (1 - CE_{x,s})]$$
$$= \sum_s [A_s \cdot EEF_{x,s}] \qquad \text{Eq. (A2)}$$

Here, $E_x$ is the total mass of emissions for species $x$, $EF_{x,s}$ is its associated emission factor for a specific source/sector $s$, and $A_s$ is the activity level of the source. $CE_{x,s}$ corresponds to effectiveness of control measure and $EEF_{x,s} = EF_{x,s} \cdot (1 - CE_{x,s})$ is the effective emission factor. When we take the ratio of emissions (Eq. A2) of co-emitted species $x$ and $y$, we obtain

$$\frac{E_y}{E_x} = \frac{\sum_s [A_s \cdot EEF_{y,s}]}{\sum_s [A_s \cdot EEF_{x,s}]}$$
$$= \sum_s \left(\frac{EEF_{y,s}}{EEF_{x,s}}\right)\left(\frac{E_{x,s}}{E_{x,total}}\right) \qquad \text{Eq. (A3)}$$

This ratio can be expressed as the sum of the products of the ratio of effective emission factors ($R^{EEF}_{x,y,s}$) and the fractional contribution of emission sector f for species ($f_{x,s}$).

**Fig. 1.** Math behind Emission Ratios

[Figure]

[Figure]

**Figure 4a.** Schematic of plume formation and transport. The $(x_p, y_p)$ and $(x_b, y_b)$ indicate mixing ratios within the fresh plume at the time of sampling and in background air in proximity to the fire, respectively. The $(x_p', y_p')$, and $(x_b', y_b')$ indicate mixing ratios within the plume sampled by the aircraft and in background air in the vicinity of the sampled plume, respectively.

**Figure 4b.** Schematic of plume sampling. Long-dashed line indicates a fresh plume with enhancements $(x_p\text{-}x_b, y_p\text{-}y_b)$ relative to background mixing ratios near the fire $(x_b, y_b)$. Dotted line indicates a plume remote from the fire which the aircraft measures relative to local background mixing ratios $(x_b', y_b')$. Enhancement ratios are obtained from the slopes of these lines.

**Fig. 2.** Mauzerall et al 1998 Figure 4 on Enhancement Ratio

$$y = \begin{bmatrix} y_1 \\ y_2 \end{bmatrix} = Hx + e = \begin{bmatrix} 1 & 0 \\ 0 & 1 \end{bmatrix}\begin{bmatrix} x_1 \\ x_2 \end{bmatrix} + \begin{bmatrix} e_1 \\ e_2 \end{bmatrix}, \quad e \sim N(0, R) \text{ where } R = \begin{bmatrix} 1 & \alpha \\ \alpha & 1 \end{bmatrix}$$

and $x \sim N(x^b, B)$, where $B = \begin{bmatrix} 1 & 0 \\ 0 & 1 \end{bmatrix}$

then, the first ($x^a$) and second ($A$) moment of the conditional distribution, $p(x|y)$ would be:

$$x^a = (H^T R^{-1} H + B^{-1})^{-1}(H^T R^{-1} H y + B^{-1} x^b), \qquad A = (H^T R^{-1} H + B^{-1})^{-1}$$

respectively. Substituting, we obtain:

$$A = \left(\frac{1}{\alpha^2 - 4}\right)\begin{pmatrix} \alpha^2 - 2 & -\alpha \\ \alpha - & \alpha^2 - 2 \end{pmatrix} = \begin{pmatrix} \frac{\alpha^2 - 2}{\alpha^2 - 4} & \frac{\alpha}{4 - \alpha^2} \\ \frac{\alpha}{4 - \alpha^2} & \frac{\alpha^2 - 2}{\alpha^2 - 4} \end{pmatrix}.$$

If we use the other expression for $A$:

$$A = B - BH^T(HBH^T + R)^{-1}HB = (I - GH)B$$

this produces the same elements of $A$, i.e.,

$$A = \begin{pmatrix} 1 - \frac{2}{4 - \alpha^2} & \frac{\alpha}{4 - \alpha^2} \\ \frac{\alpha}{4 - \alpha^2} & 1 - \frac{2}{4 - \alpha^2} \end{pmatrix}.$$

Hence, $A_{1,1} = \frac{\alpha^2 - 2}{\alpha^2 - 4} = 1 - \frac{2}{4 - \alpha^2}$ as pointed out by Peter.

Now if we let $B = \begin{bmatrix} 1 & \beta \\ \beta & 1 \end{bmatrix}$ we get:

$$A = \left(\frac{1}{\alpha^2 + \beta^2 + \alpha\beta - 4}\right)\begin{pmatrix} \alpha^2 + \beta^2 - 2 & -\alpha + \alpha\beta^2 + \alpha^2\beta - b \\ -\alpha + \alpha\beta^2 + \alpha^2\beta - b & \alpha^2 + \beta^2 - 2 \end{pmatrix}$$

**Fig. 3.** Math behind Error Correlations

$$\frac{\Delta CO}{\Delta CO_2} = \frac{cov(CO, CO_2)}{var(CO_2)} = \rho_{CO, CO_2} \frac{\sigma_{CO}}{\sigma_{CO_2}}$$

**Fig. 4.** Relationship between Enhancement Ratio and Correlation

$$x^a = (H^T R^{-1} H + B^{-1})^{-1}(H^T R^{-1} Hy + B^{-1}x^b)$$

or

$$x^a = x^b + BH^T(HBH^T + R)^{-1}(y - Hx^b)$$

to:

$$x^a - x^b = BH^T(HBH^T)^{-1}(HBH^T)(HBH^T + R)^{-1}(y - Hx^b)$$

we can easily recognize that the increment $x^a - x^b$ is a function of three terms:

1) Innovation: $(y - Hx^b)$, which is the mismatch of observed and modeled abundance
2) Least squares fit $(HBH^T)(HBH^T + R)^{-1}$ which is the ratio of the error variances
3) Linear regression: $BH^T(HBH^T)^{-1} = cov(He_x, e_y)/var(e_y)$ where $B = cov(e_y, e_y^T)$ which is the slope of a line.

In this manuscript, we highlight the need to investigate 3).

**Fig. 5.** Posterior Mean Estimate Recasted

---

## Author Comment (AC3) · 26 Feb 2021

Correction to Figure 4 (on linear regression term) of Authors' Reply to both reviewers. It should be covariance between e_x and He_x (not He_x and e_y).

---

## Author Comment (AC4) · 26 Feb 2021

We have revised the manuscript in response to reviewers comments.

—- 1) Title: We added 'potential' to the role of CO since our inversions are intended to be only demonstrative.

—- 2) Abstract: We have edited the abstract to reflect this study as a demonstration and to emphasize the global nature of our simulations which is being mainly evaluated for a given region.

—- 3) Introduction:

a) We shortened this section by removing general references on CO2 inversions. b) We also added some statements regarding the link of CO to ffCO2 and highlighted more the motivation of this study. c) We clarified our objectives and removed unnecessary text.

—- 4) Methods

2.1 We added references to EDGAR emission inventories. 2.2 We also added a clarifying statement regarding our the scope of our CAM-chem evaluation

—- 5) Results

—- Section 3: We added a header statement regarding our aim to evaluate CAM-Chem. Added some minor edits to improve clarity (also removed confusing statements).

—- Section 4: We added a header statement regarding why we use flight groups. Added some minor edits to improve clarity (also removed confusing statements).

—- Section 5: Refining ffCO2 estimates using CO and CO2 data We added a couple of equations to highlight the link of CO to ffCO2 inversion (see Figure 4 of our response to RC1 and RC2).

Added some minor edits to improve clarity –e.g., point to specific section number. Also removed confusing statements.

Added statements about covariance localization and the use of a constant 1ppm for ffCO2 offset.

Added statements on the link of CO to ffCO2 through error correlation in a priori error covariance matrix.

Noted as well that the emission inventories of CO and CO2 have been taken from different sources and that our inverse analysis focuses on CO constraints on ffCO2 total emissions (not emission ratio).

—- Section 6: Discussion and limitations

Replaced Figure 10 with a figure showing sensitivity of ffCO2 estimates to prior error correlation. Moved original Figure 10 to Figure S11.

Also added Figure S10. This is a plot of the gain matrix for non-zero error correlation and zero error correlation of a priori error covariance matrix. Deleted confusing statements.

—- 6) Conclusions: Shortened section 7.

[Figure]

**Fig. 1.** Analysis increments for East Asia and Kor+Jap ffCO2 a posteriori estimates (red) and associated error reduction (blue) as a function of CO:CO2 error correlation values in a priori error covariance (re

**Fig. 2.** Mean vertical profiles of ffCO2 response functions from Kor+Jap (blue), East Asia (red) and ROW (yellow-orange) for each flight group. Dashed and solid lines correspond to a priori and a posteriori es

[Figure]